



# Mesoscale modelling of North Sea wind resources with COSMO-CLM: model evaluation and impact assessment of future wind farm characteristics on cluster-scale wake losses

Ruben Borgers[1], Marieke Dirksen[2], Ine L. Wijnant[3], Andrew Stepek[3], Ad Stoffelen[3], Naveed Akhtar[4], Jérôme Neirynck[1], Jonas Van de Walle[1], Johan Meyers[5], and Nicole P.M. van Lipzig[1]

[1]Department of Earth and Environmental Sciences, KU Leuven, Leuven, Belgium
[2]Technical University of Delft (TUD), Delft, The Netherlands
[3]Royal Netherlands Meteorological Institute (KNMI), De Bilt, The Netherlands
[4]Institute of Coastal Systems-Analysis and Modeling, Helmholtz-Zentrum Hereon, Geesthacht, Germany
[5]Department of Mechanical Engineering, KU Leuven, Leuven, Belgium

**Correspondence:** Ruben Borgers (ruben.borgers@kuleuven.be)

**Abstract.**

As many coastal regions experience a rapid increase in offshore wind farm installations, inter-farm distances become smaller with a tendency to install larger turbines at high capacity densities. It is however not clear how the wake losses in wind farm clusters depend on the characteristics and spacing of the individual wind farms. Here, we quantify this based on multiple COSMO-CLM simulations, each of which assumes a different, spatially invariant combination of the turbine type and capacity density in a projected, future wind farm layout in the North Sea. An evaluation of the modelled wind climate with mast and lidar data for the period 2008-2020 indicates that the frequency distributions of wind speed and wind direction at turbine hub height are skillfully modelled and the seasonal and inter-annual variations in wind speed are represented well. The wind farm simulations indicate that at a capacity density of 8.1 MW km$^{-2}$ and for SW-winds, inter-farm wakes can reduce the capacity factor at the inflow edge of wind farms from 59% to between 55% and 40% depending on the proximity, size and number of the upwind farms. However, the long-term impact of wake losses in and between wind farms is mitigated by adopting next-generation, 15 MW wind turbines instead of 5 MW turbines, as the annual energy generation over all wind farms in the simulation is increased by 24% at the same capacity density. In contrast, the impact of wake losses is exacerbated with increasing capacity density, as the layout-integrated, annual capacity factor varies between 54.4% and 44.3% over the considered range of 3.5 to 10 MW km$^{-2}$. Overall, wind farm characteristics and inter-farm distances play an essential role in cluster-scale wake losses, which should be taken into account in future wind farm planning.

## 1 Introduction

The global capacity of offshore wind technologies has increased more than tenfold over the previous decade as part of the urgent transition to low-emission energy systems (IPCC, 2022). In 2021, the unprecedented commissioning of over 17 GW of offshore wind capacity pushed the cumulative, global capacity past 50 GW (Musial et al., 2022). In Europe, hosting more than





half of that global offshore capacity, annual growth rates are expected to surpass 4 GW per year in 2023 (Komusanac et al., 2021). At the same time, the size and capacity of individual turbines are increasing, with a global average rating of 7.4 MW (8.5 MW in Europe) in 2021 compared to 3.3 MW in 2011 (Komusanac et al., 2021; Musial et al., 2022). As wind turbines are organized in arrays, the total efficiency is impacted by wake effects which strongly depend on the inter-turbine spacing and

the size of the wind farm (Meyers and Meneveau, 2012; Stevens et al., 2016; Antonini and Caldeira, 2021). Currently, limited space and the urgent decarbonization of electricity systems lead to dense and large wind farms, as capacity densities can reach 10 MW km$^{-2}$ and gigawatt-scale wind farms emerge (Borrmann et al., 2018; Badger et al., 2020; Komusanac et al., 2020). On top of that, hotspots such as the North Sea are becoming more densely built (Matthijsen et al., 2018), which amplifies the risk of inter-farm interference through far-field wind farm wakes. These can extend several tens of kilometers (Platis et al., 2018;

Schneemann et al., 2020) and can lead to considerable reductions in the wind resource (Lundquist et al., 2019; Akhtar et al., 2021; Munters et al., 2022). These developments raise questions on the magnitude of intra- and inter-farm wake losses in a future, densely clustered wind farm layout including large wind farms. Mesoscale models have been applied to illustrate the strongly reduced efficiency of very large wind farms (Volker et al., 2017; Antonini and Caldeira, 2021; Pryor et al., 2021) and how this depends on the turbine spacing (Volker et al., 2017), but also how wind farms can significantly alter the energy yield

of neighbouring wind farms (Akhtar et al., 2021; Fischereit et al., 2022b). In this study, we aim to complement the existing work by quantifying how the long-term effect of wake losses in a hypothetical, future North Sea wind farm layout depends on the characteristics of the individual wind farms and on the inter-farm distances. Concretely, this is done based on a set of continuous simulations for one representative wind year, with each simulation including a different, but spatially invariant combination of the turbine type and capacity density for the wind farms in a projected, future wind farm layout.

Although the WRF model is the most commonly used mesoscale model for wind energy applications (Fischereit et al., 2022a), it is important to involve several mesoscale models to determine whether signals are robust, especially when going to climatological timescales. In this study, we make use of the regional climate model COSMO-CLM which has previously been applied for mesoscale wind farm simulations (Chatterjee et al., 2016; Akhtar et al., 2021; Akhtar et al., 2022) and also for the modelling of wind and wind resources of the past (Reyers et al., 2015; Geyer et al., 2015; Li et al., 2016) and future

(Nolan et al., 2014 ; Santos et al., 2015; Reyers et al., 2016). The quality of mesoscale wind farm simulations relies heavily on the accurate simulation of the background wind climate, which is why these models are typically evaluated with in situ, lidar and / or satellite data (Hahmann et al., 2015; van Stratum et al., 2022, Dirksen et al., 2022). The COSMO-CLM model has been shown to skilfully reproduce winds from LES (Chatterjee et al., 2016) and measurements by offshore masts (Geyer et al., 2015; Akhtar et al., 2021). However, these evaluations have only considered a limited number of datasets and time periods.

Therefore, an additional objective of this study is to extend the evaluation of COSMO-CLM based on a large set of multi-year, spatially distributed mast and wind lidar data and a satellite product covering most of the North Sea. With the focus on the wind resource, the evaluation includes metrics of power production derived from the modelled and measured wind speed data.





## 2 Data and methods

### 2.1 Model description

The development of the regional climate model COSMO-CLM (COSMO version 5.0, CLM version 15) is a joint effort between the COnsortium for Small-scale MOdelling (COSMO) and the Climate Limited-area Modelling community (CLM-Community) (Rockel et al., 2008). The Runge-Kutta dynamical core solves the non-hydrostatic, compressible hydro-thermodynamical equations on a rotated latitude-longitude grid (Doms and Baldauf, 2013). Several coordinates are possible in the vertical dimension, of which we used the height-based, terrain-following coordinate with grid stretching. Additional physical processes

are parametrized in the model: for subgrid-scale turbulence the standard choice was adopted, which is the one-dimensional diagnostic closure scheme (level 2.5) which is based on a prognostic TKE equation after Mellor and Yamada (1982) (Raschendorfer, 2001). Surface fluxes are also parametrized and are coupled to the included multi-layer soil model, TERRA-ML. Also parametrizations for grid-scale clouds and precipitation, moist convection and radiative processes are included (Doms et al., 2013). An extensive description of the model system is available in the documentation (e.g. Doms and Baldauf, 2013).

The simulation domain covers a large fraction of the North Sea with a horizontal grid spacing of 0.025° ($\sim$ 2.8 km) (Fig. 1). In the vertical dimension, 61 levels were used up to an elevation of 22 km with a spacing of approximately 20 meters near the surface and 30 meters at turbine hub height. The relaxation zone at the lateral boundaries is 40 km wide and transitions into a spin-up zone that is an additional 100 km in width, in agreement with the recommendations of Brisson et al. (2015). Omission of this 140 km-wide zone gives the evaluation domain (Fig. 1). The ERA5 reanalysis (Hersbach et al., 2020) was

used as forcing at the boundaries with updates every hour. No additional nesting stages were used, again in agreement with Brisson et al. (2015). As deep convection is explicitly resolved at the meso-$\gamma$ scale (Vergara-Temprado et al., 2020), only shallow convection was parametrized according to Tiedtke (1989). In COSMO5.0, the TKE advection term in the prognostic equation is only included for the experimental, LES-type turbulence schemes. With the focus on wind farm wake development in the second part of this study, we implemented the TKE advection term for the standard turbulence scheme in COSMO5.0

based on version 5.01.

Specific to this study, we also employed the Fitch wind farm parametrization (Fitch et al., 2012), that has been implemented in COSMO5-CLM15 (Chatterjee et al., 2016; Akhtar and Chatterjee, 2020). This additional module represents the wind farm forcing on the atmosphere as a sink of kinetic energy and a source of TKE. A downward correction for the TKE coefficient has been proposed based on a comparison with large eddy simulations (LES) (Archer et al., 2020), while other studies do not

necessarily find that this leads to better performance (Siedersleben et al., 2020; Larsén and Fischereit, 2021), hence the original value was retained in this study. The farm wakes generated by the Fitch parametrization have been evaluated in COSMO-CLM using LES Chatterjee et al., 2016 and aircraft measurements Akhtar et al., 2021 and also in WRF and HARMONIE using offshore masts and wind lidars in the vicinity of wind farms (Garcia-Santiago et al., 2022; Dirksen et al., 2022). In addition, the validation of HARMONIE showed that employing the Fitch WFP also led to skilful simulation of wind speed statistics inside

of a wind farm (Dirksen et al., 2022). For a detailed overview of the performance validation of this parametrization, we refer to the review of Fischereit et al. (2022a).





**Figure 1.** Map of the study area showing the simulation domain (cyan, solid line) and the evaluation domain (cyan, dashed line). Also the locations of the in situ measurement stations (orange dots) and lidar stations (green triangles) that are used for the model evaluation are indicated, in addition to the hypothetical future wind farm layout (grey polygons) and the four analysis transects TR1-TR4 (yellow lines) used for the wind farm simulations. Map created using QGIS3.4.

## 2.2 Evaluation run

To evaluate the model performance, a simulation was performed for a period of 13 years (2008-2020) to ensure that the inter-annual variability in the wind field and wind energy conditions were sampled well (Geyer et al., 2015; Ronda et al., 2017).
Measurements from in situ, lidar and satellite data over the North Sea are abundant in both space and time over this period. The wind farm parametrization was excluded in this simulation since a time-static wind farm layout cannot represent the rapidly growing wind farm layout over this time period and most observations were representative for wind farm-free conditions. Hence, only the undisturbed wind climate was evaluated and the observations were filtered accordingly (cf. section 2.4.1). The





instantaneous wind field around hub height was written to output at a 10 minute frequency following the standard for wind

energy assessments (Menezes et al., 2020).

## 2.3 Wind farm simulations

The projected future wind farm layout used in the wind farm simulations was constructed from the EMODNET wind farm dataset (EMODnet, 2022) and GIS data from the Royal Belgian Institute for Natural Sciences (Vigin, 2022) (Fig. 1). Next to the operational wind farms today, this layout incorporates the concessions that are in different stages of the constructions

process, zones for which consent has been authorised and also large development zones. Wind turbines were assumed constantly operational, unless the wind speed was below the cut-in wind speed or above the cut-out wind speed. Considering the computational cost of these experiments, the timespan was limited to one representative year in terms of the North Sea wind field. This year was determined in a procedure based on the one outlined in Tammelin et al. (2013). We used 31 years of hourly, hub-height wind fields from the ERA5 reanalysis (1990-2020), to compute a metric R for the representativeness per year and

per grid cell:

$$R_{i,j,y} = \frac{S1_{i,j,y}}{\sigma_{S1}} + \frac{S2_{i,j,y}}{\sigma_{S2}} + \frac{S3_{i,j,y}}{\sigma_{S3}} \qquad (1)$$

where the indices $i$, $j$ and $y$ refers to a specific grid cell and year. These R values were computed per year for each North Sea grid cell between 51° and 55.5° latitude. Higher values of R correspond to more representative years. The different scores ($S1$-$S3$) are based on the agreement between single-year and the long-term (31 year) histograms as computed by the Perkins

Skill Score:

$$PSS(H_1, H_2) = \sum_{b=1}^{n} MIN(F_{H_1}^b, F_{H_2}^b) \qquad (2)$$

where $H_1$ and $H_2$ represent the first and second histogram and $F^b$ represents the normalized frequency for bin b. A PSS of 1 (or 100%) represents complete overlap of two histograms. $S1$ is the PSS between a wind speed histogram for a single year and the multi-year wind speed histogram, using a bin width of 0.5 m/s. $S2$ is the same as $S1$, but for wind direction,

using a bin width of 30°. Finally, $S3$ represents the mean PSS between the single- and multi-year wind speed distributions over 12 wind direction sectors. The scores ($S1$-$S3$) are standardized by the standard deviation to give each term in the sum equal weight. Summation of $R$ over all grid cells then yields a representativeness for a specific year. The different scores and the final score per year are summarized in supplementary figures S1 and S2, respectively. Based on this procedure, the simulations were carried out for the year 2016, as the representativeness is high overall, but especially for the wind direction distribution,

which is particularly important in this work.

Five simulations were performed, consisting of one simulation without any wind farms (NOWF) and four simulations using a fixed wind farm lay-out with the same turbine type and capacity density for all wind farms (Table 1). Based on the number of turbines, the total capacity and the surface area of operational wind farms in the North Sea, a median turbine capacity of





4.85 MW and a representative capacity density of 8.1 MW km$^{-2}$ were determined. The 5 MW reference wind turbine of the National Renewable Energy Laboratory (NREL) (Jonkman et al., 2009)) with a hub height of 90 m and a rotor diameter of 126 m was therefore used in conjunction with the aforementioned capacity density in one of the wind farm simulations (NREL8.1). Three additional cases were simulated in which the NREL 5 MW was replaced by the 15 MW reference wind turbine of the International Energy Agency (IEA) (Gaertner et al., 2020) with a hub height of 150 m and a rotor diameter of 240 m, as 15 MW turbines are expected to reach the market in a few years and are now being selected for upcoming projects (Bento and

Fontes, 2019; Shields et al., 2021). The power curves of these three turbines are available in supplementary Fig. S3. The three cases with 15 MW turbines were simulated with a different wind farm capacity density:

- IEA3.5: low capacity density in which the inter-turbine distance is 10 rotor diameters. This turbine spacing is larger than is found in most offshore wind farms today and corresponds to a lower cost per unit energy production as the impact of turbine wakes is reduced and is most relevant in regions where offshore space is relatively abundant, such as for the
United Kingdom or Denmark (Borrmann et al., 2018).

- IEA8.1: The same capacity density as for the NREL8.1 scenario.

- IEA10.0: high capacity density with a larger revenue per unit area, but also increased wake-related losses. This corresponds to a capacity density for planned projects in regions where the available space is limited, such as Belgium, Netherlands and Germany (Borrmann et al., 2018).

**Table 1.** Summary of the turbine type and capacity density used in the different wind farm model simulations.

| Identifier | turbine type | capacity density (MW/km$^2$) |
|---|---|---|
| NOWF | / | / |
| NREL8.1 | NREL 5 MW | 8.1 |
| IEA3.5 | IEA 15 MW | 3.5 |
| IEA8.1 | IEA 15 MW | 8.1 |
| IEA10.0 | IEA 15 MW | 10 |

Based on the different simulations, the impact of the turbine type and capacity density on the wake losses was assessed. In addition, the roles of wind farm size and inter-farm distance in these wake losses were investigated based on the large variation in both over the wind farm layout. The different simulations were compared along the transects indicated on Fig. 1, which correspond to dominant, but also strongly disturbed wind directions, i.e. directions along which the wind farms are densely clustered. For this analysis, only winds in a sector of 30° around the transect orientation (SW to NE for TR1, TR2 and TR4 and

NW to SE for TR3) were selected based on the center grid cell on the transect. The data selection based on the wind direction reduced the dataset to approximately 14% of the total for transects TR1, TR2 and TR4 and to 8.1% for TR3. Additionally,





this transect analysis was extended to three stability classes based on the Bulk Richardson Number (BRN), a metric for the dynamic stability (cf. section 2.5.3).

## 2.4 Measurement data

### 150 2.4.1 in situ masts

Wind measurements of 19 in situ stations (Fig. 1) were obtained from the KNMI data platform, Meetnet Vlaamse Banken, the Marine Data Exchange, the FINO data platform and the TNO wind energy data platform (Table A1). Of these 19 stations, 6 were actual meteorological masts with measuring devices at multiple altitudes. The remaining stations correspond to coastal measurement poles and instrumentation mounted on oil-, gas- or light-platforms and provide information at a single altitude.

Average wind speed and wind direction are available at 10 minute intervals. A timeline of the data availability is summarized for each station in supplementary Fig. S4.

For most stations, corrections were applied to the measurements of the boom- or platform-mounted anemometers and wind vanes in order to account for flow distortions by the mast or other mounting infrastructure. These corrections were performed by the data providers for the stations FINO1 and FINO3 (Westerhellweg et al., 2012; Leiding et al., 2016), MMIJ (Werkhoven

and Verhoef, 2012), WH and WA. For the remaining stations with multiple anemometers per height level, we avoided using measurements in the wake of the mast or other infrastructure by selecting the measurement with the highest 10 minute average wind speed. A possible drawback of this approach is that the measured wind speed is overestimated in the case of lateral speed-up effects (Leiding et al., 2016). If wind direction was provided with respect to magnetic north, a magnetic-to-true north correction was applied according to the location and timing of the dataset. Finally, since no wind farm parametrization was

included in the evaluation run, measurements potentially taken in the wake of wind farms were omitted from the dataset by filtering out either a specific time range or a directional sector. These dataset corrections are summarized in supplementary Table S1. A station-to-farm distance threshold of 50 km was chosen to perform these corrections, as it is expected that the impact of wind farm wakes on the long-term wind speed statistics becomes relatively unimportant at this distance (Schneemann et al., 2020; Dirksen et al., 2022). The total uncertainty on the wind speed measurements is a combination of the uncertainties of

calibration, mounting (incl. flow obstruction by the mast), data acquisition and the local site conditions. This total uncertainty can vary significantly between the stations. For the class 0.9A anemometers at the station MMIJ the total uncertainty was estimated at 1.5% for the top anemometer and 1.9% for the boom-mounted anemometers (Duncan et al., 2019). For the top anemometers of the other meteorological masts, which have a comparable class number as for MMIJ (Friis Pedersen et al., 2006), we applied the same value of 1.5% as the uncertainty estimate. As the boom-mounted anemometers at the FINO stations

were also mast-corrected prior to use, we adopted the same value of 1.9%. The mounting uncertainty for boom-anemometers at the stations GG, LA and HGW is expected to be larger since we only performed a simple correction. Assuming an additional 2% uncertainty on the mast correction, this leads to a total uncertainty of 3.7%. For the remaining stations, we assumed a calibration uncertainty of 1.5% (Coquilla et al., 2007), an operational uncertainty of 0.8% (Friis Pedersen et al., 2006) and an augmented 2% uncertainty on the data acquisition due to limited information on acquisition and post-processing. For AWG1,



P11B and WH a mounting uncertainty of 5% was estimated due to presence of lateral flow obstructions. For the other stations, where the device is mounted on the top of a platform or platform-mounted mast, a mounting uncertainty of 2% was assumed following Verkaik (2001).

### 2.4.2 Wind lidar

In addition to the cup anemometers, measurements from 6 wind lidars were used for the evaluation (Fig. 1). These lidars
use light beam scanning technology to derive vertical profiles of wind speed and direction at regular height intervals and allow evaluation of the wind field above the typical 90 m top of meteorological masts. As for the in situ measurements, wind speed and direction is provided as 10 minute averages. The data were obtained from the Dutch services TNO wind energy and Rijksdienst voor Ondernemend Nederland (RVO). The lidars were installed during the pre-construction stages of offshore wind farm development (Table A2). The LEGO, MMIJ, K13 and EPL lidars are installed on the same platforms as the cup
anemometers (Table A1). The BO and TNW lidars are floating lidars and are mounted on a Fugro SEAWATCH buoy. Estimates of the uncertainty are from Wouters et al. (2019a, 2019b, 2019c) for LEGO, EPL and K13, from Poveda and Wouters (2015) for MMIJ and from the report by Dhirendra (2014) for the floating lidars BO and TNW.

### 2.4.3 ASCAT

The Advanced SCATterometer (ASCAT) sensor on the European MetOp satellites uses radar technology to determine the
near-surface wind speed and direction over the sea (Gelsthorpe et al., 2000; Figa-Saldaña et al., 2002). While the ASCAT product only provides information on the surface wind, it complements the in situ and lidar data as it covers most of the North Sea basin. For this study, we considered the L3-reprocessed ascending and descending passes of the MetOp-A satellite from the the website of the Copernicus Marine Service (CMEMS). The satellite was operational for the complete 13 years of this simulation. Specifically, the variant on a 12.5 km grid, with a horizontal resolution of 25 km was used, which has been validated
against buoy measurments (Verhoef and Stoffelen, 2009). The long-term instrumental stability is estimated to be below 0.1 m/s for this product, whereas the climatological uncertainty is ±0.1 m/s, with some anomalies of +1 m/s at the Dutch coast. The datasets for both passes together provide roughly one instantaneous measurement per day for most of the North Sea that we consider (4500 samples in total). Only close to the coasts, data coverage is much lower (100-3000 samples), which is a well-known issue with remotely sensed winds related to contamination with land signal (Bourassa et al., 2019).

## 2.5 Evaluation approach

### 2.5.1 Model collocation with in situ and lidar

Over a 10 minute period, the wind travels over a distance comparable to the edge length of a 0.025° grid cell. Since the model wind components represent smoothed grid box averages, the 10 minute time-averages of the observations were directly compared to instantaneous values of the grid cell in which the station is located. In the case of gaps in the time-series of the in



situ and lidar data, the corresponding timestamps were also eliminated from the model grid point time-series. The model wind speed data was extrapolated to the measurement heights using the wind profile power law:

$$V_s = V(h_m) \cdot (\frac{h_s}{h_m})^\alpha \tag{3}$$

where $V_s$ is the wind speed at sensor height, $V(h_m)$ is the wind speed at the first model level below sensor height and $\alpha$ is the shear coefficient which is computed as:

$$\alpha = \frac{\ln(V(h_{m+1}) \,/\, V(h_m))}{\ln(h_{m+1} \,/\, h_m)} \tag{4}$$

where $m+1$ denotes the first model level above sensor height. In contrast to the wind speed, the model wind direction at sensor height was computed after linear interpolation of the horizontal wind components of the model levels just above and below sensor height.

The Zephir 300S lidar has a well-known 180° ambiguity that can occur in the wind direction time-series as it relies on a sonic anemometer just above the lidar to determine the sign of the wind vector. In the case of low wind speeds and/or flow obstructions, it is possible that the incorrect sign is determined and the lidar's wind direction is 180° off (Knoop et al., 2020). We corrected this 180° error by adding or subtracting 180° if the wind direction in the measurements differs more than 90° from the modelled wind direction (∼2% occurrence) after Dirksen et al. (2022).

### 2.5.2  Model collocation with ASCAT and triple collocation

For the comparison with ASCAT, the model surface winds were regridded to the 12.5 km grid of the measurements with first-order conservative remapping. This ensures that all the source grid cells contained within a target grid cell have similar weight in the regridding, in agreement with the ASCAT winds being computed from the signal of this complete area. Afterwards, the measurement time-series of each ASCAT grid cell was matched by a model time-series for that same grid cell by linear interpolation in time.

Additionally, a comparison between the model, ASCAT and in situ data was conducted at the stations WH, EPL and MMIJ. These stations were selected because the location is far enough from the coast to ensure sufficient data points in the ASCAT data and the measurement height is close to 10 m, which reduces any vertical extrapolation errors to 10 m in the in situ data. This extrapolation was done using the power law with a constant shear coefficient of 0.11. The in situ data was then also linearly interpolated to the ASCAT measurement times and all datasets were limited to the timings where both ASCAT and in situ measurements are available. Finally, the grid cells in which the stations are located were selected from the model and ASCAT datasets for the comparison.



### 2.5.3 Stability classification

The comparison between COSMO-CLM and the measurements in terms of wind speeds was further extended to different classes of atmospheric, dynamic stability since this strongly determines the wind conditions over the North Sea (Stull, 1988; Sathe et al., 2011) and also determines the atmospheric response to a wind farm forcing (Platis et al., 2021). This stability classification was done based on the Bulk Richardson Number (BRN), which is computed as:

$$BRN = \frac{\frac{g}{\theta_v} \frac{\Delta \theta_v}{\Delta z}}{(\frac{\Delta u}{\Delta z})^2 + (\frac{\Delta v}{\Delta z})^2} \tag{5}$$

where $g$ corresponds to the gravitational constant, $\theta_v$ is the virtual potential temperature, $z$ is height and $u$ and $v$ are the zonal and meridional wind speed components, respectively. The overbar over the virtual potential temperature denotes that it is averaged over the different model layers between 50 m and 150 m height. Finally, the gradients in $u$, $v$ and $\theta_v$ were determined by averaging the gradients between each of the subsequent layers between 50 m and 150 m. Based on the BRN, we can identify three distinct dynamic stability regimes (Grachev et al., 2013; Dirksen et al., 2022):

- – unstable: BRN < 0. This is the case when the temperature gradient is negative, which corresponds to an unstable thermal stratification.

- – weakly stable: 0 < BRN < 0.25. This is the case when the temperature gradient is positive, but the temperature effect is weak compared to the vertical wind shear. In this case, the wind shear-generated turbulence is relatively strong compared to the buoyant damping.

- – stable: BRN > 0.25. This is the case when the temperature gradient is positive and strong compared to the vertical shear. In this case, the wind shear-generated turbulence is strongly damped and this region of the ABL can be considered dynamically stable.

Gradients were calculated based on potential temperature instead of virtual potential temperature as an analysis of the driving data showed minimal variations of specific humidity over the considered height range. Since vertical profiles of pressure and temperature are generally not available over the range of the meteorological masts or wind lidar scanning ranges, this can only be computed for the model. For a comparison with the measurement data, the corresponding timestamps for each stability class in the model data were also selected in the measurement time series.

### 2.5.4 Evaluation metrics

We compared the magnitudes of the mean wind speed difference and the observational uncertainty to identify any model bias: an exceedance of the observational uncertainty at a measurement station, was used as the threshold for the presence of a model bias at that location. In addition, the PSS (cf. section 2.3) was employed as a metric to express the agreement in the shape of two histograms of either wind speed or wind direction.





Since the relationship between wind speed and wind turbine power production is nonlinear, we also evaluated differences between COSMO-CLM and the observations in terms of the capacity factor, which is given by:

$$CF = 100 \cdot \frac{\sum_{i=1}^{n} P(V_i) \cdot \Delta t}{n \cdot P(V_r) \cdot \Delta t} \quad [\%] \tag{6}$$

where $V_i$ is the hub-height wind speed at some instance $i$ in the time-series, $\Delta t$ is the time-interval between points in the
series, $V_r$ is the rated wind speed and P is the generated power which depends on the wind speed and is turbine-specific. So, the capacity factor is the ratio between the energy production of a specific turbine based on a wind speed time-series and the theoretical, maximum energy production over that same period, i.e. for a turbine continuously operating at full capacity. This is an idealised notion of the capacity factor as it concerns an isolated turbine which constantly operates according to the power curve. For these calculations, we considered the power curve of the NREL 5 MW reference wind turbine, with a hub height of
90 m, for the meteorological masts with the top anemometer below 100 m and the power curve of the DTU 10 MW reference wind turbine (Bak et al., 2013), with a hub height of 119 m, for FINO1, FINO3 and the wind lidars (supplementary Fig. S3). An uncertainty range on the capacity factor based on the observed wind speeds was determined based on the uncertainty on the wind speed measurements: the observed wind speed distribution was shifted linearly by the product of the uncertainty and the mean wind speed after which upper and lower bounds on the capacity factor were computed. As the capacity factor is a
percentage, absolute differences are also a percentage so to avoid confusion it is always explicitly stated whether absolute or relative differences in the capacity factor are considered.

## 3 Results and discussion

### 3.1 Evaluation

#### 3.1.1 Wind speed up to 290 m

The difference in the mean wind speed between the in situ and lidar stations varies with height (Fig. 2). Below 90 meters, the difference is generally negative (model underestimates the mean) and exceeds the measurement uncertainty range, indicating a bias. However, the magnitude of the bias generally drops with increasing elevation over this height range, albeit with some exceptions (MMIJ, TNW). At measurement heights at or above 90 meters, the difference is generally smaller and falls within the uncertainty range of the measurements. The gradient with height persists and the difference is positive above 130 meters
at the locations of the wind lidars. While differences over height are substantial, there is no robust indication of regional differences in the ability of COSMO-CLM to model the climatological mean wind speed. The same figure but with relative differences is included in the supplementary materials (supplementary Fig. S5).





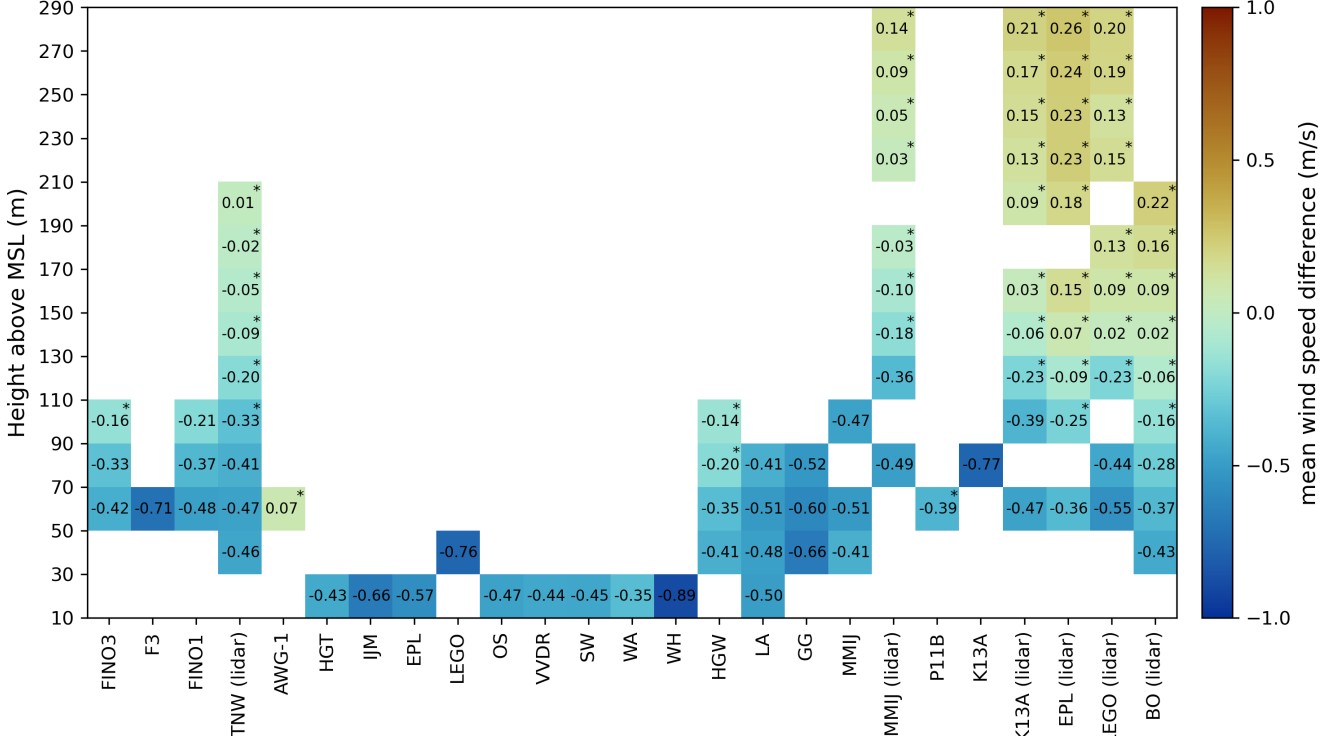

**Figure 2.** Wind speed bias (m/s) for the complete time period of the measurement data. This concerns measurements between 10 and 290 meters MSL. The vertical range is subdivided into 20 meter intervals for readability. The presence of an asterisk indicates that the bias is within the measurement uncertainty. Stations are clustered per region.

The magnitude of the mean difference with ASCAT is below 0.5 m/s for most grid cells (Fig. 3). For approximately 45% of the grid cells the mean difference is within the ASCAT climatological uncertainty of ±0.1 m/s. These grid cells are generally located further from the coast and correspond to the regions without in situ measurements, which is an indication of good model performance in this region. The near-surface bias identified against the in situ data in the southern North Sea (Fig. 2) is much smaller than the differences compared to ASCAT in this region. A three-way comparison with three in situ stations shows that the mean differences against the in situ data exceed the in situ measurement uncertainty for both COSMO-CLM and ASCAT. Where the model generally underestimates the mean near-surface wind speed, ASCAT overestimates it with a larger magnitude, which explains the PSS values. The PSS scores are similar when both COSMO-CLM and ASCAT are corrected for the systematic bias with respect to the in situ data, which indicates that both perform similarly in approximating the distribution shape of the in situ data.





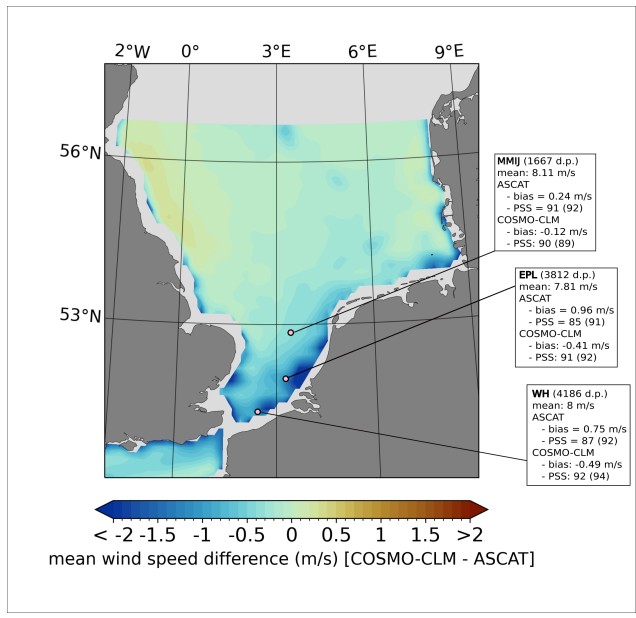

**Figure 3.** Difference in long-term mean wind speed between COSMO-CLM and ASCAT. Yellow dots indicate the measurement stations for triple collocation. The text boxes summarizes the mean 10 meter wind speed for three in-situ stations and the agreement of ASCAT and CCLM in terms of the mean difference and the PSS. PSS values between brackets are after elimination of the mean difference between the two histograms to remove the effect of distribution location.

### 3.1.2 Wind statistics and power metrics at hub height

The distributions of wind speed near 100 meters height match well with the meteorological masts and lidar stations in most
cases (Fig. 4), leading to a PSS generally above 95%. The associated absolute differences in the idealized capacity factor are within the uncertainty on the capacity factor based on the wind speed measurements for 4 out of 10 stations. For FINO1, K13, MMIJ and HGW the differences are outside the uncertainty range, but the deviations from the lower bound of the uncertainty range are less than 1%, while the deviations are higher for the GG and LA masts. For K13 and HGW the capacity factor difference exceeds the uncertainty based on the measurements, whereas the mean wind speed difference is within the
measurement uncertainty which can be linked to the non-linear relationship between wind speed and power production.





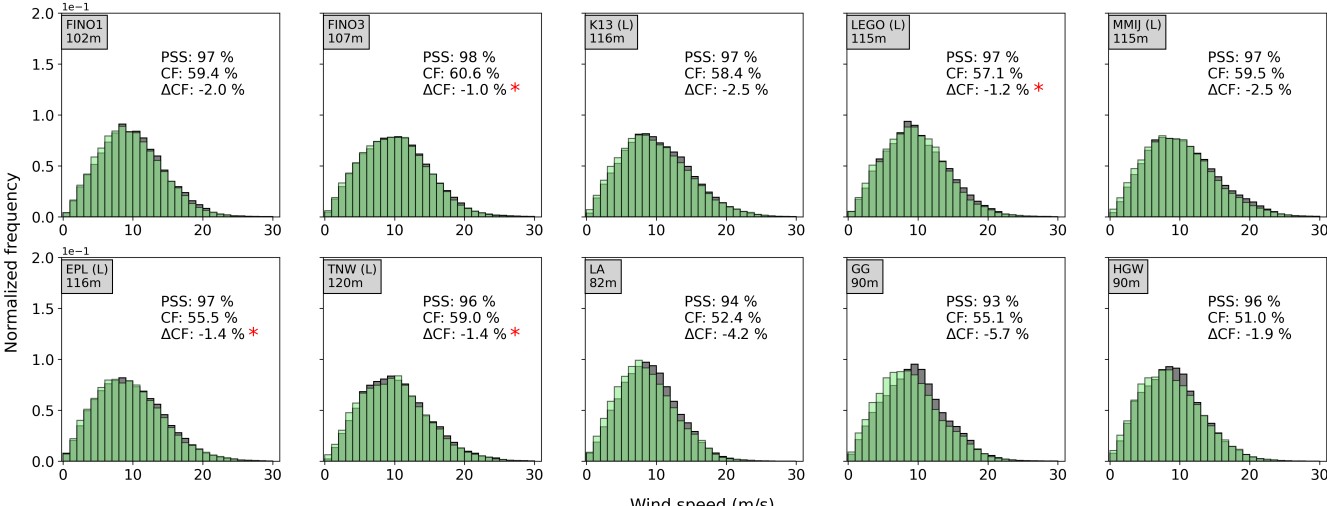

**Figure 4.** Histograms of the collocated wind speed datasets: measurements (in grey) and the model (in green). In addition, the associated PSS, the capacity factor based on the measured wind speeds and the absolute difference in capacity factor between the model and the measurements are indicated. The presence of a red asterisk indicates that the capacity factor difference falls within the uncertainty on the capacity factor for the measurements.

While the inter-annual variability of the annual mean hub-height wind speed is typically around 1 m/s, the corresponding variability in the wind speed bias between the model and the measurements is typically around 0.1 m/s or 10% of that value (cf. Table 2). The corresponding overlap between the single-year histograms generally does not vary with more than 2% over the years. Hence, the agreement in distribution location and shape between COSMO-CLM and the measurements remains
relatively stable over consecutive years, regardless of the inter-annual variability in the wind conditions.

**Table 2.** Inter-annual range of the mean wind speed and of the agreement between the model and observations, as expressed by the min/max annual bias and the min/max annual PSS for the different years in the measurement period.

| Station | Period (NR years) | Annual mean (m/s) | | bias (m/s) | | PSS (%) | |
|---|---|---|---|---|---|---|---|
| | | MAX | MIN | MAX | MIN | MAX | MIN |
| FINO3 (107m) | 2010-2013 (4) | 11.2 | 9.5 | -0.11 | -0.22 | 98 | 97 |
| MMIJ (92m) | 2012-2015 (4) | 10.3 | 9.8 | -0.44 | -0.50 | 96 | 94 |
| K13 lidar (115m) | 2018-2020 (3) | 10.4 | 9.9 | -0.2 | -0.28 | 97 | 97 |
| LEGO lidar (115m) | 2015-2020 (6) | 11.1 | 9.2 | -0.18 | -0.28 | 97 | 95 |
| LA (82m) | 2008-2010 (3) | 9.5 | 8.6 | -0.32 | -0.47 | 95 | 93 |



Also the intra-annual cycle in the wind speed distribution is well represented by the model (Fig. 5). The gradual seasonal variation from higher (winter) to lower (summer) median wind speeds is accurately reproduced in addition to the variation in distribution width (Q25-Q75 range) and more extreme conditions (Q5 and Q95). Moreover, also in extreme months the model succeeds in modelling the wind speed distribution as can be deduced from Fig. 5b at station TNW for February of 2020, albeit

with a heavier right tail and consequently more winds above the cut-out wind speed.

**Figure 5.** Boxplots representing the multi-year wind speed distribution per month for the observations (black) and the model (red). Shown for the three masts and three lidar stations at turbine hub height. The box corresponds to the Q25-Q50-Q75 wind speeds. The lower and upper whisker are the Q5 and the Q95 percentiles, respectively.

Evaluation of the long-term wind direction histograms near turbine hub height (using a bin width of 20°) shows an overlap of 95% or more in most cases (Table 3) with the magnitude of the bias generally below 4°. A reason for the stronger deviation at FINO3 and EPL has not been identified. Also the variations of the wind speed statistics with the wind direction are captured by





the model (supplementary Fig. S6). This accurate reproduction of the wind direction distributions and the direction-dependent

wind speed distributions is encouraging for the application to wind farm modelling as wind farm shapes are tailored to the

regional wind climate.

**Table 3.** Bias in the wind direction [model - observations] and the Perkins Skill Score between the histograms of wind direction (bin width = 20°).

| station | bias (°) | PSS (%) |
|---|---|---|
| FINO3 (101 m) | -8.0 | 96 |
| FINO1 (91 m) | 1.9 | 95 |
| TNW lidar (120 m) | -4.0 | 97 |
| K13A lidar (116 m) | -2.2 | 97 |
| Ijmuiden lidar (115 m) | 1.0 | 96 |
| Europlatform lidar (116 m) | 8.7 | 93 |
| LEGO lidar (115 m) | 0.7 | 96 |
| London Array (78 m) | -1.9 | 96 |
| Humber Gateway (86 m) | 2.3 | 96 |
| Greater Gabbard (62 m) | -3.5 | 97 |

### 3.1.3 Dynamic stability

The general differences in mean wind speed profiles for the three stability classes agree well between the model and the measurements (Fig. 6): winds are strongest under weakly stable conditions and weakest under stable condition with the wind speeds

under unstable conditions falling in between. The agreement between the profiles of the model and the measurements differs

between the stability classes: under stable conditions the shear in the model is too strong between 40 and 200 meters leading

to a negative model bias below 160 m for EPL and LEGO and below 180 m for K13 and TNW. Around 100 m, the respective underestimations are at least 0.3 m/s and 0.6 m/s. Such an underestimation under stable conditions is not uncommon for

climate models (Wijnant et al., 2014; Sheridan et al., 2021). For weakly stable conditions, there is not a clear bias around 100

meters, but the deviations below 90 m and above 150 m are outside of the observational uncertainty. The small vertical gradient

under unstable conditions is represented well by the model with only small deviations that are well within the measurement

uncertainty over the complete height range. The hub-height wind speed distributions as reflected in the boxplot mainly differ

in distribution location with the strongest differences under stable conditions. Corresponding capacity factor values were cal-

culated with lower and upper uncertainty bounds for the observations (supplementary Fig. S7). Under stable conditions, the

deviations between model and observations exceed the uncertainty range, hence the absolute model underestimation of the

capacity factor is at least 2.5%. For unstable and weakly stable conditions, the deviations are within the uncertainty range of

the observations.



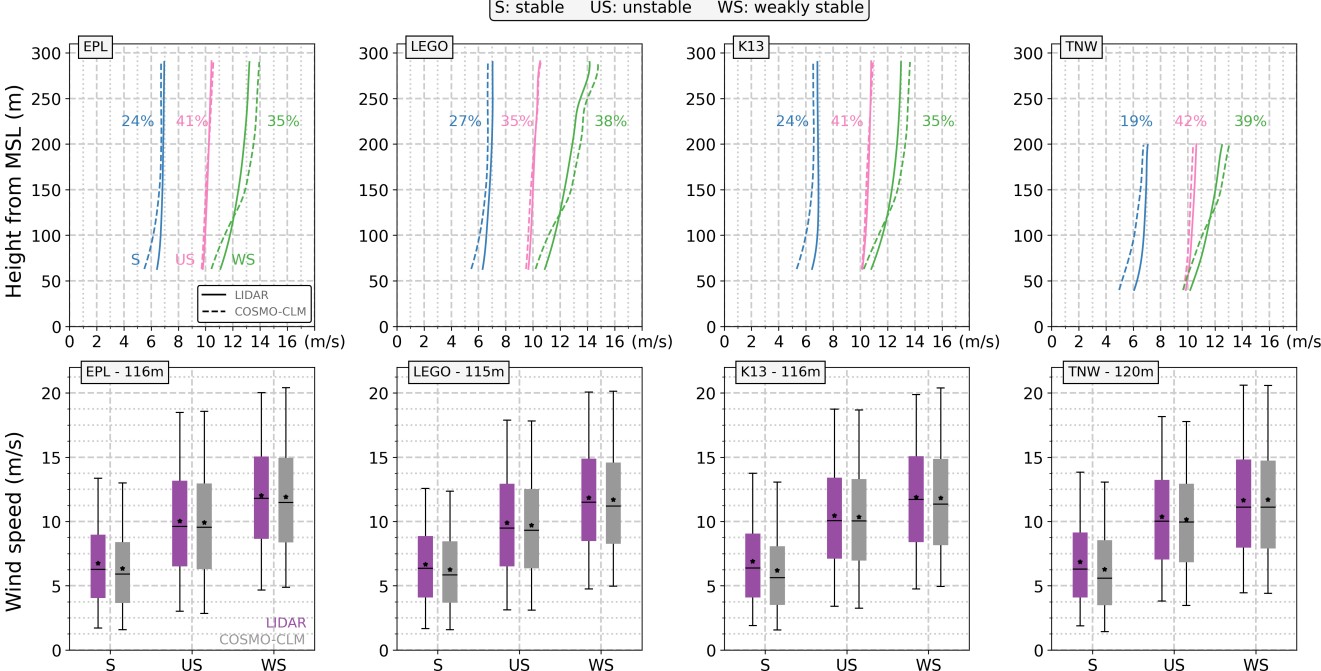

**Figure 6.** Model evaluation for different stability classes. Top row: vertical profiles of the mean wind speed per stability class for 4 lidar stations (full line) and the corresponding model output (dashed line). The stability classes are stable (blue), unstable (pink), weakly stable (green). The indicated percentages are the relative frequency of the different stability classes at hub height. Bottom row: boxplots of the hub height wind speeds per stability class for the same 4 lidar stations (purple) and the corresponding model output (grey). The box corresponds to the Q25-Q50-Q75 wind speeds. The asterisk indicates the mean and the lower and upper whisker are the Q5 and the Q95 percentiles, respectively.

## 3.2 Effect of wind farm characteristics

The modelled mean wind speed at 90 m for 2016 varies from 7.5 m/s at the coast up to 10 m/s in the open North Sea (Fig. 7).
The associated capacity factor varies between 45% and 60% and the simulated pattern agrees well with earlier, multi-decadal estimates over the North Sea (Geyer et al., 2015). Stability separation shows that the the capacity factors are generally largest under weakly stable conditions and can reach 75% in the open North Sea. For stable conditions capacity factors are considerably lower, but also prone to the bias discussed in section 3.1.3. The bottom row of Fig. 7 visualizes the impact of the projected, future wind farm layout would they be all occupied with NREL 5 MW turbines at 8.1 MW km$^{-2}$. Without subdividing for
stability, the absolute reductions of the full-year capacity factor in the vicinity of farms located in dense clusters can reach 10%, with cumulative contributions from multiple wind farms. The magnitude of the long-term resource reductions are similar to what other studies have identified in terms of closely spaced wind farms (Akhtar et al., 2021; Fischereit et al., 2022b). Very close to the larger farms, larger values can be found, even when the farms are isolated. The absolute and relative changes in the capacity factor vary over the stability classes. Absolute capacity factor reductions are typically the smallest for stable



conditions, but these are the largest in relative terms as capacity factors are small themselves. In weakly stable conditions, absolute reductions are much higher with deficits exceeding 4% more than 20 km from wind farm clusters and larger wind farms and exceeding 10% over large zones within and outside the wind farm clusters.

**Figure 7.** Maps of the modelled North Sea wind climatology, the corresponding wind resource in terms of the capacity factor and the resource deficit under the NREL8.1 scenario for the complete year and for the three stability classes. Top: Maps of the yearly mean wind speed (m/s) under the NOWF scenario. Middle: capacity factor under the NOWF scenario (%). Bottom: absolute capacity factor deficit for the NREL8.1 scenario (%). White polygons represent wind farms. Capacity factor computations are based on the power curve of the NREL 5 MW wind turbine.

The impact of the atmospheric stability on the wind farm-induced reduction in hub-height wind speed can be analysed in more detail along the four analysis transects (Fig. 8). For TR1, TR2 and TR4, the data is dominated by weakly stable conditions



(∼800 data points) compared to unstable (∼240 data points) and stable (∼200 data points) conditions, while for TR3 unstable

conditions are more prevalent (∼400 data points) compared to stable (∼200 data points) and weakly stable (∼80 data points)

conditions. The reductions under an unstable atmosphere are generally smaller at the downwind edge of some wind farms than

for stable and weakly stable conditions, for which maxima can exceed 30%. As a result, regions of densely clustered wind

farms (inter-farm distance < 20 km) are characterized by more intense, relative deficits in the mean wind speed under (weakly)

stable conditions than under unstable conditions. Translating this into capacity factor profiles shows that the relative impact on

the wind resource is much larger for stable conditions than for weakly stable and unstable conditions (supplementary Fig. S8).

This is because wind speeds under stable conditions are frequently within the steep part of the power curve (cf. Fig. 6), which

results in a larger effect on the capacity factor for downwind grid cells compared to when a larger fraction of the wind speed

distribution is above the rated wind speed.

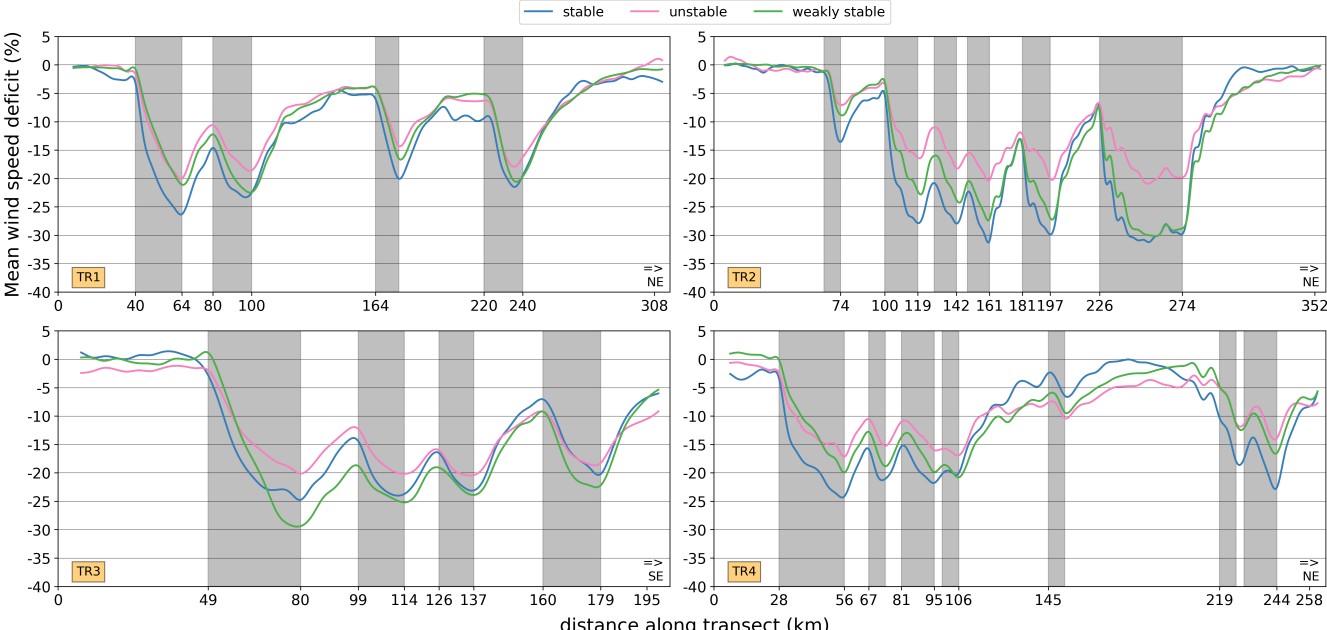

**Figure 8.** Relative deficit of the along-transect mean wind speed (%) for the four transects indicated on Fig. 1. This concerns the NREL8.1
scenario, subdivided in the three dynamic stability classes: unstable (pink), weakly stable (green) and stable (blue) according to the BRN.
Wind data is only considered when the wind direction deviates within +-15° from the transect orientation (W to E) at the middle grid cell of
each transect. Grey shadings represent wind farms.

The wind farm capacity density used in the different wind farm simulations strongly determines the mean wind speed profile

along these transects (Fig. 9). In each case, zones of densely clustered farms (< 20 km apart) are characterized by the strongest

reductions and a limited wake recovery that typically does not exceed half of the maximum reduction. The scenario with IEA

15 MW turbines at 3.5 MW km$^{-2}$ (blue) is characterized by the smallest reductions, which are typically within 1 m/s at the

upwind side of wind farms. For higher capacity densities, these upwind reductions are often more than twice as large and can





exceed 2 m/s depending on the degree of clustering. Only for recovery distances larger than 50 km, the IEA8.1 and IEA10.0 scenarios converge to within 0.5 m/s of the IEA3.5 scenario. Furthermore, the impact of wind farm size on the intensity of the reduction can be assessed by focusing on the first wind farm in each transect: the larger wind farms of TR1, TR3, TR4 have an along-transect farm length between 24 km and 31 km, while this is only 9 km for the one in TR2. The associated reductions at the downwind edge of the wind farms (NREL8.1) are approximately twice as large for TR1, TR3 and TR4 than for TR2.

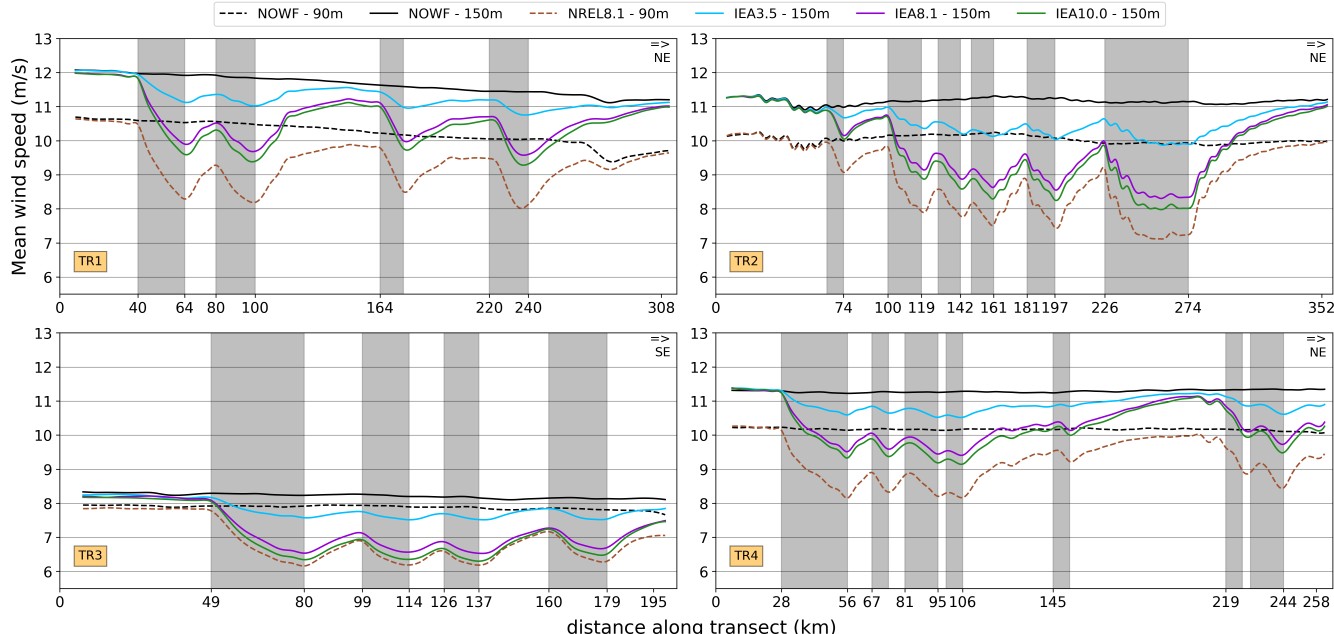

**Figure 9.** Transects of the mean wind speed for the different wind farm scenarios. These transects correspond to the four lines in Fig. 1. Wind data is only considered when the wind direction deviates within +-15° from the transect orientation (W to E) at the middle grid cell of each transect. Grey shadings represent wind farms.

When converting the wind speed information of the NOWF scenario into capacity factors, the transect-averages are ∼59% for TR1, TR2 and TR4 and ∼39% for TR3 when considering the hub height and power curve of the NREL 5 MW turbine. For the IEA 15 MW turbine, these increase to ∼67% and ∼47%, respectively. Fig. 10 shows that the associated, absolute reductions in these capacity factor follow the general patterns established for the mean wind speed. In each transect, the IEA3.5 scenario (blue) is characterized by the smallest deficits at the upwind edge of wind farms, typically around 5% with larger values in

dense clusters. For higher capacity densities, the upwind edge reductions reach 15% to 20% for closely spaced wind farms. The intensity of these upwind edge reductions are strongly dependent on the degree of upwind clustering and the sizes of the upwind farms. For the scenarios wit higher capacity densities, the stronger upwind edge reductions are heavily exacerbated throughout the farms as the capacity factor declines steeply, eventually resulting in much lower wind farm efficiencies in these situations. For the SW-NE oriented transects, the impact of the turbine type becomes apparent: for the 90 m turbines in the

NREL8.1 scenario, the absolute deficits over the wind farms exceed those of the IEA8.1 scenario, which translates to a much



stronger reduction in relative terms as the unaltered (NOWF) capacity factors for the 5 MW turbines are lower than for the 15 MW turbines.

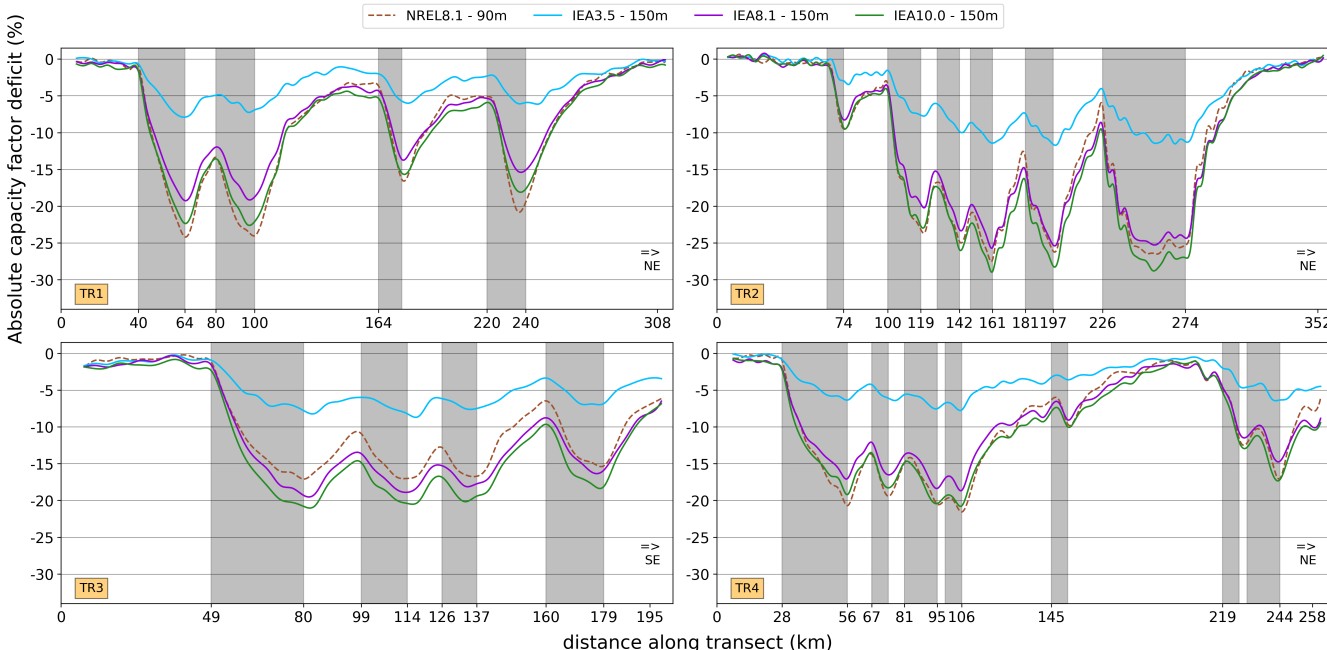

**Figure 10.** Transects of the absolute capacity factor deficit for the different wind farm scenarios. These transects correspond to the four lines in Fig. 1. Wind data is only considered when the wind direction deviates within +-15° from the transect orientation (W to E) at the middle grid cell of each transect. Grey shadings represent wind farms.

The wind farm layout in the IEA8.1 scenario is significantly more efficient than for the NREL8.1 scenario, as reflected in the integrated capacity factor and full load hours over all wind farms in the evaluation domain (Table 4). As a consequence, the

integrated, annual energy generation is 24% higher in the former. This difference is partly due to the rated wind speed being 0.8 m/s lower for the 15 MW turbines so that the rated section of the power curve is more wide (cf. supplementary Fig. S2). Added to that, taller turbines can take advantage of the wind speed gradient with height, which leads to a larger fraction of wind speeds in the rated regime and a reduced fraction in the steep part of the power curve. To disentangle both effects, the 90 m wind speed data of the NREL8.1 scenario was fed to the 15 MW power curve, which resulted in a total generation of 740

TWh. This implies that approximately 46% of the increase in total generation can be attributed to the lower rated wind speed and approximately 54% to the wind speed gradient with height.

Combining 15 MW turbines with a low capacity density of 3.5 MW km$^{-2}$ only reduces the integrated capacity factor from 60.1% in the NOWF scenario to 54.4%, as a consequence of limited intra- and inter-farm wake impacts, in agreement with (Meyers and Meneveau, 2012) and (Gupta and Baidya Roy, 2021). From IEA3.5 to IEA8.1 the capacity density increases by

131%, whereas the total generation only increases by 96%. From IEA8.1 to IEA10.0 these increases are 23.4% and 18.4%,



respectively. This efficiency degradation when moving to larger capacity densities can be recognized in a reduced capacity factor and a reduction in the full load hours (FLH): compared to IEA3.5, the IEA10.0 capacity factor is reduced from 54.4% to 44.3% and the FLH is reduced by approximately 850 hours. This follows from the increased wake losses that are further exacerbated by the densely clustered layout and the presence of several large wind farms that are typically characterized by

very low power densities (Volker et al., 2017).

**Table 4.** Annual energy and power metrics integrated over all wind farms in the evaluation domain. CF: layout-integrated capacity factor. FLH: full load hours for the complete layout. The calculations are based on the wind speed data of the wind farm grid cells. The capacity factors for the NOWF simulation correspond to efficiency in absence of intra- and inter-farm wakes.

| experiment | turbine | total capacity (GW) | CF (%) | FLH (h) | total generation (TWh) |
| --- | --- | --- | --- | --- | --- |
| NOWF | NREL 5 MW | / | 52.3 | / | / |
| NOWF | IEA 15 MW | / | 60.1 | / | / |
| NREL8.1 | NREL 5 MW | 205 | 37.3 | 3261 | 668 |
| IEA3.5 | IEA 15 MW | 89 | 54.4 | 4726 | 421 |
| IEA8.1 | IEA 15 MW | 205 | 46.1 | 4028 | 826 |
| IEA10.0 | IEA 15 MW | 253 | 44.3 | 3868 | 979 |

## 4    Conclusions

We have used the regional climate model COSMO-CLM to quantify the dependence of long-term, cluster-scale wake losses on the turbine model, capacity density, wind farm spacing and wind farm size for a hypothetical future wind farm layout consisting of operational wind farms, construction-phase wind farms and development zones. First, the model skill in simulating the wind

climate was evaluated in a comparison with in situ, lidar and satellite data, which revealed a negative model bias near the surface, but not at turbine hub height for most stations. Furthermore, the overlap between the long-term hub-height wind speed and wind direction distributions is generally above 95% and both seasonal, inter-annual and directional variability of the wind speed distributions at turbine hub height are captured well. Separation into stability classes reveals small, height-dependent biases under dynamically stable conditions. The capacity factors derived from the modelled and observed wind speed data

agree well overall, but small, long-term underestimations are present at some locations. As deviations mainly occur under stable conditions, a stability-dependent bias correction could be considered for future applications in addition to continuous efforts to improve the model. Overall, this evaluation emphasizes the value of having a large set of wind measurements available in regions for offshore wind farm development, as it allows a benchmarking of mesoscale models over the region of interest. The application of the model to a hypothetical, future wind farm layout indicates that the creation of dense wind farm clusters

is accompanied by an alteration of the surrounding wind climate and significant farm-to-farm wake interactions. The impact of these interactions depend heavily on the turbine model, the capacity density, the inter-farm spacing and the size of the wind



farms. In this study, the comparison of two turbine types (NREL 5 MW, IEA 15 MW) and three capacity densities (3.5, 8.1 and 10 MW km$^{-2}$) show that:

- For a capacity density of 8.1 MW km$^{-2}$, the basin-integrated total generation is 24% larger for a layout of 15 MW turbines than for 5 MW turbines and the relative impact of wake effects is smaller. This can be explained by the increase of the wind resource with height (54%) and a wider rated section in the power curve for the IEA 15 MW turbine (46%).

- Under dominant wind directions with dense wind farm clustering, the wind resource is strongly reduced due to inter-farm wakes. Assuming 15 MW turbines, the absolute reductions of the capacity factor at the upwind edge of wind farms range from 3% to 8% for a capacity density of 3.5 MW km$^{-2}$ depending on the degree of clustering and the size of the upwind farms. For a capacity density of 8.1 MW km$^{-2}$ this ranges from 5% to 20% and for 10 MW km$^{-2}$ from 5% to 23%.

- Assuming a projected, future wind farm layout with 15 MW turbines, increases of the capacity density of the wind farms lead to strong efficiency reductions. The layout-integrated capacity factor is 10% lower for a 10 MW km$^{-2}$ capacity density than for a 3.5 MW km$^{-2}$ capacity density, due to the intensification of intra- and inter-farm wake losses.

- The relative impact of wake effects on the wind resource is strongest under stable conditions with a BRN larger than 0.25, as the wind speeds largely coincide with the steep ramp of the power curve. These are the lowest resource periods for which a halving of the capacity factor is possible under strongly disturbed wind directions. While these results are possibly impacted by the negative model bias that was found for stable stratification, it is expected that the general tendency towards a stronger relative impact under stable conditions still holds.

It is important to consider that these simulations within the wind farm do not consider subgrid-scale wake effects, but assume all the turbines in one grid cell operate at the same inflow wind speed. While this is generally not the case, it is difficult to quantify the net effect of this assumption on the parameterized momentum deficit, as the turbine-induced wind speed reductions are a non-linear function of the wind speed itself, with a maximum reduction at the rated wind speed. While the mesoscale wind farm parametrization approach has limitations, these modelling studies provide valuable information for the efficient deployment and operation of offshore wind infrastructure, more so because climate models can consider the climatic variability of wake effects, for large regions. This study demonstrated the potential of this modelling approach to explore a large variety of wind farm characteristics and layouts in a climatic context, which can aid in a more efficient expansion of the offshore infrastructure.

*Code and data availability.*

The code and data used to generate figures 3-10 can be retrieved as one dataset at https://doi.org/10.5281/zenodo.7767102. The ERA5 reanalysis data used to identify representative wind years were downloaded via the Copernicus Climate Change Service (C3S) Climate Data Store (CDS) and can be found at https://doi.org/10.24381/cds.adbb2d47. The ASCAT data was





retrieved from the Copernicus Marine Service via https://doi.org/10.48670/moi-00183. The in-situ measurements of the KNMI can be retrieved from their data platform, at https://dataplatform.knmi.nl/. For the in situ data at the Belgian coast, data is accessible via the website of the Belgian coastal measurement network, at https://meetnetvlaamsebanken.be/. Mast data at the coast of the United Kingdom are available via the website of the Marine Data Exchange, at https://www.marinedataexchange.co.uk/ and for the German Bight via the website of the FINO dataplatform, at http://fino.bsh.de/. Data from the Ijmuiden meteorological mast and from the platform-mounted wind lidars can be found at the TNO data cloud website https://nimbus.windopzee.net/ and the data from the buoy-mounted lidars at https://offshorewind.rvo.nl/.

**Appendix A**





**Table A1.** Description of the in situ measurement stations. S: wind speed (m/s), D: wind from direction (°). The superscripts a, b and c link measurement heights to measurement devices in the next column. 1x, 2x and 3x refers to one, two or three anemometers and/or wind vanes at one measurement height. Source acronyms: KDP: Royal Netherlands Meteorological Institute (KNMI) Data Platform, MNVB: Meetnet Vlaamse Banken, MDE: the Marine Data Exchange, FINO: Forschungsplattformen in Nord- und Ostsee, TNO: Nederlandse Organisatie voor Toegepast-natuurwetenschappelijk Onderzoek

| Name (abbreviation) | location | heights (m MSL) | measured variables | period | uncertainty (%) | source |
|---|---|---|---|---|---|---|
| Westhinder (WH) | platform | 26 | 2x S,D | 2008 - 2020 | 5.6 | MNVB |
| Wandelaar (WA) | measuring pole | 26 | 2x S, 1x D | 2013 - 2020 | 3.3 | MNVB |
| Scheur-Wielingen (SW) | measuring pole | 25 | 1x S,D | 2010 - 2020 | 3.3 | MNVB |
| Oosterschelde (OS) | measuring pole | 17 | | 2008 - June 2019 | 3.3 | KDP |
| Vlakte van de Raan (VVDR) | measuring pole | 17 | | Sep. 2009 - June 2019 | 3.3 | KDP |
| Lichteiland Goeree (LEGO) | platform | 38 | 2x S,D | 2008 - 2020 | 3.3 | KDP |
| Europlatform (EPL) | platform | 29 | 2x S,D | 2008 - 2020 | 3.3 | KDP |
| Ijmond (IJM) | measuring pole | 17 | | 2008 - June 2019 | 3.3 | KDP |
| P11-B (P11B) | mast on platform | 51 | 2x S,D | 2010 - 2020 | 5.6 | KDP |
| Meteomast Ijmuiden (MMIJ) | meteorological mast | $27,58^a$ , $92^b$ | a: 3x S,D, b: 2x S | Nov. 2011 - Mar. 2016 | a: 1.9; b: 1.5 | TNO |
| K13A (K13) | mast on platform | 74 | 2x S,D | 2008 - 2019 | 3.3 | KDP |
| F3N (F3) | mast on platform | 60 | 2x S,D | 2010 - Dec. 2019 | 3.3 | KDP |
| Huibertgat (HGT) | measuring pole | 18 | | 2008 - June 2019 | 3.3 | KDP |
| AWG-1 (AWG1) | mast on platform | 60 | 2x S,D | Sep. 2009 - 2020 | 5.6 | KDP |
| FINO1 (FINO1) | meteorological mast | $51,71,91^a$ , $102^b$ | a: 1x S,D; b: 1x S | 2008 - July 2009 | a: 1.9; b: 1.5 | FINO |
| FINO3 (FINO3) | meteorological mast | $50,70,90,100^a$ , $107^b$ | a: 3x S,D; b: 1x S | 2009 - Oct. 2014 | a: 1.9; b: 1.5 | FINO |
| Humber Gateway (HGW) | meteorological mast | $34,52,70,88^a$ , $68^b$ , $90^c$ | a: 2x S; b: 1x D; c: 1x S | Oct. 2009 - Jul. 2011 | a: 3.7; c: 1.5 | MDE |
| Greater Gabbard (GG) | meteorological mast | $42,52,72,82^a$ , $62^b$ , $88^c$ | a: 2x S; b: 2x D; c: 1x S | 2008 - 2010 | a: 3.7; c: 1.5 | MDE |
| London Array (LA) | meteorological mast | $20,32,57^a$ , $29,78^b$ , $82^c$ | a: 2x S; b: 1x D; c: 1x S | 2008 - 2010 | a: 3.7; c: 1.5 | MDE |



**Table A2.** Description of the lidar measurement stations. Source acronyms: RVO: Rijksdienst voor Ondernemend Nederland, TNO: Nederlandse Organisatie voor Toegepast-natuurwetenschappelijk Onderzoek

| Name (abbreviation) | Type | Location | heights (m MSL) | Period | uncertainty (%) | Source |
|---|---|---|---|---|---|---|
| Borssele 1 (BO) | Zephir 300S | buoy | 40:20:200 | Jun. 2015 - Feb. 2017 | 3.3 - 3.4 | RVO |
| Lichteiland Goeree (LEGO) | Leosphere windcube | platform | 90:25:290 and 63 | Nov. 2014 - 2020 | 2.6 - 3.3 | TNO |
| Europlatform (EPL) | Zephir 300S | platform | 91:25:291 and 63 | May 2016 - 2020 | 2.9 - 3.5 | TNO |
| Meteomast Ijmuiden (MMIJ) | Zephir 300S | platform | 90:25:290 | Nov. 2011 - Mar. 2016 | 2.5 - 3.1 | TNO |
| K13A (K13) | Zephir 300S | platform | 91:25:291 and 63 | 2018 - 2020 | 2.7 - 3.2 | TNO |
| TNVD Waddeneilanden A (TNW) | Zephir 300S | buoy | 40:20:200 | Sep. 2019 - 2020 | 3.3 - 3.4 | RVO |





*Author contributions.*

Ruben Borgers contributed to the conceptualization, data curation, formal analysis, investigation, methodology, project administration, software, validation, visualization and writing (original draft, review and editing). Marieke Dirksen contributed to the data curation, resources, methodology and writing (review and editing). Ine Wijnant, Andrew Stepek and Ad Stoffelen contributed to the resources, methodology and writing (review and editing). Naveed Akhtar contributed to the methodology,
software and writing (review and editing). Jérôme Neirynck and Jonas Van de Walle contributed to the methodology and writing (review and editing). Johan Meyers contributed to the conceptualization, funding acquisition, methodology, project administration, supervision and writing (review and editing). Nicole P.M. van Lipzig contributed to the conceptualization, funding acquisition, investigation, methodology, project administration, resources, supervision, visualization and writing (original draft, review and editing).

*Competing interests.*  One of the co-authors, Johan Meyers, is an associate editor of the WES journal. The peer-review process was guided by an independent editor, and the authors have also no other competing interests to declare.

*Acknowledgements.*  The authors acknowledge support from the project FREEWIND, funded by the Energy Transition Fund of the Belgian Federal Public Service for Economy, SMEs, and Energy (FOD Economie, K.M.O., Middenstand en Energie). The computational resources and services in this work were provided by the VSC (Flemish Supercomputer Center), funded by the Research Foundation Flanders (FWO)
and the Flemish Government department EWI.

The authors further acknowledge the COSMO-CLM community for the support in the modelling efforts done in this study. Also EUMETSAT OSI-SAF is acknowledged, in which Ad Stoffelen is involved.

The authors further thank the Royal Netherlands meteorological institute (KNMI), the Meetnet Vlaamse Banken, the German Federal Maritime And Hydrographic Agency (BSH), the Marine Data Exchange (MDE) for the in situ wind measurements, metadata and additional data
handling support. Also Energieonderzoekcentrum Nederland (ECN) and the Nederlandse Organisatie voor toegepast-natuurwetenschappelijk onderzoek (TNO) are thanked for the mast and lidar measurements at IJmuiden and lidar measurements at Lichteiland Goeree, Europlatform and K13A, and Rijksdienst voor Ondernemend Nederland (RVO) is thanked for the lidar data of Borssele and Ten Noorden van de Waddeneilanden. Finally, Copernicus and the Copernicus Marine Service are acknowledged for the ERA5 reanalysis and MetOp-A ASCAT measurements.



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
