# Peer review of "Mesoscale modelling of North Sea wind resources with COSMO-CLM: model evaluation and impact assessment of future wind farm characteristics on cluster-scale wake losses"

_Wind Energy Science, 2023_

## Referee Comment (RC1)

"Mesoscale modelling of North Sea wind resources with COSMO-CLM: model evaluation and impact assessment of future wind farm characteristics on cluster-scale wake losses" by Borgers et al.

Review by David M. Schultz

This is an important paper because it shows that the increase in height of the turbines from 5 MW to 15 MW more than offsets the loss due to wakes. I approach this review as a meteorologist and mesoscale modeler. Hence, my comments are mostly focused on these aspects of the manuscript. I see no problem with the science and the results, only the presentation.

1.  Does the paper address relevant scientific questions within the scope of WES?
    Yes.

2.  Does the paper present novel concepts, ideas, tools, or data?
    Yes.

3.  Is the paper of broad international interest?
    Yes.

4.  Are clear objectives and/or hypotheses put forward?
    Yes, very clear.

5.  Are the scientific methods valid and clear outlined to be reproduced?
    Yes, very clearly described.

6.  Are analyses and assumptions valid?
    Yes.

7.  Are the presented results sufficient to support the interpretations and associated discussion?
    Yes, they are.

8.  Is the discussion relevant and backed up?
    Yes, the discussion, which is interspersed throughout section 3, is relevant and defensible.

9.  Are accurate conclusions reached based on the presented results and discussion?
    Yes.

10. Do the authors give proper credit to related and relevant work and clearly indicate their own original contribution?
    Yes, they do.

11. Does the title clearly reflect the contents of the paper and is it informative?
    The title is quite appropriate and detailed. Yes, it is informative.

12. Does the abstract provide a concise and complete summary, including quantitative results?
    The abstract is well written.

13. Is the overall presentation well structured?
    In general, yes. I have some suggestions below about different ways to structure section 3 that would make it more readable. More signposting of the structure within

section 3 is needed, along with that renumbering of the subsections and sub subsections.

14. Is the paper written concisely and to the point?
    Yes.

15. Is the language fluent, precise, and grammatically correct?
    For the most part, yes. I have a few suggestions for the authors, enumerated below.

16. Are the figures and tables useful and all necessary?
    I think so. I didn't see that the supplemental figures were all that necessary, but I don't see the harm in including them.

17. Are mathematical formulae, symbols, abbreviations, and units correctly defined and used according to the author guidelines?
    Mostly. I have enumerated a few issues below to consider.

18. Should any parts of the paper (text, formulae, figures, tables) be clarified, reduced, combined, or eliminated?
    Mostly, no. The paper is mostly fine the way it is. I have some suggestions that will improve the readability of the figures and the text.

19. Are the number and quality of references appropriate?
    Yes. Some care should be taken with the citations, as better scholarship there would benefit the readability of the manuscript. See below.

20. Is the amount and quality of supplementary material appropriate and of added value?
    I don't see that it adds that much value, but it is appropriate. Other than changes to the figures, I don't see that it needs to be changed.

Major comments:

1. The conclusion section is unbalanced. Lines 414–423 represent an unacceptably short, incomplete, and qualitative summary of the first part of the study. In contrast, lines 424–443 represent a much more detailed and quantitative set of conclusions. I prefer the latter, as I imagine most readers would who would want to read the conclusion to get a more clear picture of the results of this study. I suggest a revision to the first part of the conclusion section.

2. Figure 2: The color scale needs work. First, the zero point should be white, not bluish-yellow, to indicate its true neutrality. It's hard to interpret otherwise. Second, the color scheme is not symmetric. Negative values are all shades of blue, but positive values are yellow, orange, and red. Instead, all positive values should be shades of red, opposite of the negative values.

In any case, I recommend to the authors to avoid the rainbow color scheme. It distorts gradients, among other issues.

Stauffer, R., G. J. Mayr, M. Dabernig, and A. Zeileis, 2015: Somewhere Over the Rainbow: How to Make Effective Use of Colors in Meteorological Visualizations. *Bull. Amer. Meteor. Soc.*, **96**, 203–216, https://doi.org/10.1175/BAMS-D-13-00155.1.

Please fix similar problems with Figures 3 and S5.

3. An excessive number of grid lines appears on Figures 5, 6, S1–S4, and S6–S9. These are distracting from the data (i.e., what Edward Tufte calls *chartjunk*) and should be eliminated.

https://www.edwardtufte.com/bboard/q-and-a-fetch-msg?msg_id=00040Z

https://en.wikipedia.org/wiki/Chartjunk

4. Model output and observations have different colors in each figure where they are compared directly against each other: Figures 4, 5, and 6 (and comparable figures in the supplement). Can a similar color scheme (red for model and black for observations, for example) be employed across all three of these figures? It would sure simplify things for the reader having that consistency across all the figures.

5. Lines 282–284: I find three levels of organization (section 3 to subsection 3.1 to subsection 3.1.1) without any text introducing each confusing. The authors need to put these sections into context before diving down three levels to a very specific quantity (e.g., wind speed at 290 m). For example, after the title for section 3, there should at least be a paragraph explaining how this section is structured and any general information that pertains to it. Also, after the title for section 3.1, there should at least be a paragraph explaining what will be discussed in this subsection and any general information that pertains to it. The same is true for section 3.2. The text just jumps right in with results from Figure 7. Can you provide some context to the reader first?

6. Line 284: Are three levels of organization necessary? Usually sections and subsections are sufficient. Could you just have different sections for results? Section 3.1 becomes section 3, section 3.2 becomes section 4, etc.? Alternatively, you could just drop the third level of subsubsections. I'm not sure they are helping the reader. It would make the text more readable and less tedious. Also, the text of section 3 is just a little over 100 lines. Three levels of subsubsections seems excessive.

Minor comments:

1. Lines 13–15: "In contrast, the impact of wake losses is exacerbated with increasing capacity density, as the layout-integrated, annual capacity factor varies between 54.4% and 44.3% over the considered range of 3.5 to 10 MW km$^{-2}$." I have read this sentence multiple times, and I am having trouble making sense of it. I think it is phrase "impact of wake losses is exacerbated" that is the problem. Could it be written more clearly?

2. Lines 15–16: "wind farm characteristics and inter-farm distances play an essential role in cluster-scale wake losses, which should be taken into account in future wind farm planning." This sentence is too vague and general to be a meaningful conclusion to your paper. For example, this sentence could be the conclusion of just about every paper in *Wind Energy Science*. It does not do your study justice.

3. Lines 25, 33: Why these three citations of all the citations that have been published on the efficiency of wind-farm wakes? In such instances, it would help to precede the list of citations with "e.g." to indicate that these are a sampling of all the possible sources that could have been cited. You may wish to consider adding "e.g." to other lists of citations, as well.

4. Line 64: In contrast, the "e.g." is not needed here because presumably there is only source for "documentation".

5. Line 47: should be "and/or", with no spaces.

6. Line 67: The verb tense changes back and forth from present to past tense in this paragraph. I think past tense sounds better, but whatever you pick, aim to be consistent.

7. Line 71: I think more careful wording of "deep convection is explicitly resolved at the meso-gamma scale" is needed.  Convective storms may start to be resolved at these scales, but the updrafts comprising that convection are not, as has been shown in Bryan et al. (2003).  So, the phrase "deep convection is explicitly resolved" is ambiguous.

Bryan, G. H., J. C. Wyngaard, and J. M. Fritsch, 2003: Resolution Requirements for the Simulation of Deep Moist Convection. Mon. Wea. Rev., 131, 2394–2416, https://doi.org/10.1175/1520-0493(2003)131<2394:RRFTSO>2.0.CO;2.

8. Lines 80, 340: "Hence" cannot be used as a conjunction in this context. https://langeek.co/en/grammar/course/752/since-vs-hence

9. Line 89: I'm unclear what role these two citations are supporting.  Period of 13 years?  Need for sampling inter-annual variability?  Other studies doing model evaluation?  More explanation is needed for why their citation is relevant to this sentence.  That may involve rewriting the sentence so that the reason for the citations becomes more clear.

10. Lines 93, 264, 367, 396: "Cf." means "compare".  So what is being compared to section 2.4.1?  It isn't clear.  Write instead "cf. A and B" to compare A to B.

11. Line 99: Change "constructions" to "construction".

12. Line 104: Delete the comma.

13. Lines 107, 112, 213, 216, 243, 269, 402, 424: Please insert \noindent before "where".

14. Lines 104 and elsewhere:  Italicize R, as it is a variable.

15. Line 112: Italicize b.

16. Lines 109, 116, and elsewhere: Change hyphens to en dashes (i.e., two hyphens in LaTeX) to connect two items in a range.  Fix also in Tables A1 and A2.  Fix throughout the manuscript.

17. Line 118:  Capitalize "Figures".

18. Lines 118–120: I am having a hard time understanding this sentence.  I think there is too much being communicated within.  Break it up, perhaps.

19. Lines 118–120: On what basis can it be said that "the representativeness is high overall, but especially for the wind direction distribution,"?  Please evidence that statement.

20. Lines 157, 219, 230: Is this a new paragraph? If so, indent it. If not, then combine it with the previous line.

21. Line 164: Change "since" to "because" to avoid implication of time that "since" implies. See also lines 207, 239, 257, 266.

22. Line 170: spell out "including".

23. Lines 195, 311, 445, 447: Change "while" to "although" to avoid the implication of simultaneity that "while" implies.

24. Line 199: Change "resolution" to "grid spacing" for consistency and precision.

25. Line 208 and throughout the manuscript: Hyphenate "10-minute period". Fix throughout.

26. Lines 214, 241, 267: Delete the colon.

27. Line 270: Italicize P.

28. Line 295: "further" should be "farther". https://www.dictionary.com/e/farther-vs-further/

29. Line 297–299: This sentence should cite Fig. 3. You were talking about Figure 3, but then cited Figure 2. You should return to citing Figure 3 to avoid any confusion and point the reader to the right figure.

30. Figure 3: If the top part of the graph is gray (i.e., presumably no difference field plotted there), then why not trim it off? Also, does it make sense to make the graph encompass the same domain as Figure 7? If so, that would be helpful to the readers to compare figures that have the exact same map background.

31. Figure 4: I am having a hard time understanding this graph. There are three colors (light green, dark green, and gray), yet only two colors are explained in the figure caption. Can you write the figure caption more clearly (or replot the graph) to make it easier to understand? I think the histograms are overlapping, but this is difficult to interpret.

32. Line 315: Is "stable" the right word? "Consistent" would be better.

33. Line 343: The title of this subsection is vague: "characteristics". It could be more clear what characteristics are being varied.

34. Line 347: Put a comma between "conditions capacity".

35. Line 360: Change "while" to "whereas" to avoid implication of simultaneity.

36. Line 364: Why is "weakly" in parentheses? You don't include the parentheses when that term is used in line 366. I suggest deleting the parentheses.

37. Figure 8 caption: Use the ± symbol.

38. Lines 373, 384: Delete "(blue)" as that information belongs in the legend and figure caption, not the text.

39. Line 387: Change "wit" to "with".

40. Line 404: Be careful of the difference between \cite and \citep.

41. Line 404: Insert a comma after "IEA8.1".

42. Line 405: Insert a comma after "IEA10.0".

43. Lines 414–415: "negative bias" in "wind climate" is unclear. Can you just say that the "model was underpredicting the wind speed"? That is easier to understand. Look for other similar opportunities throughout the manuscript to convey your message more simply and more clearly.

44. Line 417: A comma is needed after "95%" to join the two independent clauses.

45. Line 417: Delete "both" because you describe three things: "seasonal, inter-annual, and directional variability".

46. Lines 418–419: This sentence is unclear. Also, what about biases in other stability conditions?

---

## Author Comment (AC1)

**Response to the comments of David Schultz for WES-2023-33**

June 18, 2023

**General feedback from the referee**

This is an important paper because it shows that the increase in height of the turbines from 5 MW to 15 MW more than offsets the loss due to wakes. I approach this review as a meteorologist and mesoscale modeler. Hence, my comments are mostly focused on these aspects of the manuscript. I see no problem with the science and the results, only the presentation.

**General reply**

The authors thank the referee for pointing out several adaptations to improve the presentation of our results and for the suggested improvements of the text body. These suggestions will be taken up in a revision of the manuscript. We have formulated a response to each of the major comments and also to several subsets of the minor comments.

**Major comments**

**Comment 1**: The conclusion section is unbalanced. Lines 414–423 represent an unacceptably short, incomplete, and qualitative summary of the first part of the study. In contrast, lines 424–443 represent a much more detailed and quantitative set of conclusions. I prefer the latter, as I imagine most readers would who would want to read the conclusion to get a more clear picture of the results of this study. I suggest a revision to the first part of the conclusion section.

**Reply**: we recognize the imbalance in the conclusion. For the revised version, we will provide more detail and quantitative information with respect to the first part of the study.

**Comment 2**: Figure 2: The color scale needs work. First, the zero point should be white, not bluish-yellow, to indicate its true neutrality. It's hard to interpret otherwise. Second, the color scheme is not symmetric. Negative values are all shades of blue, but positive values are yellow, orange, and red. Instead, all positive values should be shades of red, opposite of the negative values. In any case, I recommend to the authors to avoid the rainbow color scheme. It distorts gradients, among other issues. Please fix similar problems with Figures 3 and S5.

**Reply**: We agree that better color scales can be used with a more clear zero-point and better symmetry to make the figures easier to interpret. Another colorblind-friendly diverging color scale will be selected for Fig. 2, Fig. 3, Fig. 5S.

**Comment 3**: An excessive number of grid lines appears on Figures 5, 6, S1–S4, and S6–S9. These are distracting from the data (i.e., what Edward Tufte calls chartjunk) and should be eliminated.

**Reply**: We recognize the excessive number of gridlines for most of the mentioned figures. For Figures 5,6, S1-S3, S6 and S7, the number of grid lines will be significantly reduced towards a later version. For Figures S8-S9, we would prefer to keep the gridlines as they are in order to aid the reader in reading the values from the y-axis.

**Comment 4**: Model output and observations have different colors in each figure where they are compared directly against each other: Figures 4, 5, and 6 (and comparable figures in the supplement). Can a similar color scheme (red for model and black for observations, for example) be employed across all three of these figures? It would sure simplify things for the reader having that consistency across all the figures.

**Reply**: Consistency is indeed lacking here and there is room for improvement. For Fig. 4, Fig. 5 and the bottom row of Fig. 6, the same set of colors will be used for the model and the observations.

**Comment 5**: Lines 282–284: I find three levels of organization (section 3 to subsection 3.1 to subsection 3.1.1) without any text introducing each confusing. The authors need to put these sections into context before diving down three levels to a very specific quantity (e.g., wind speed at 290 m). For example, after the title for section 3, there should at least be a paragraph explaining how this section is structured and any general information that pertains to it. Also, after the title for section 3.1, there should at least be a paragraph explaining what will be discussed in this subsection and any general information that pertains to it. The same is true for section 3.2. The text just jumps right in with results from Figure 7. Can you provide some context to the reader first?

**Comment 6**: Line 284: Are three levels of organization necessary? Usually sections and subsections are sufficient. Could you just have different sections for results? Section 3.1 becomes section 3, section 3.2 becomes section 4, etc.? Alternatively, you could just drop the third level of subsubsections. I'm not sure they are helping the reader. It would make the text more readable and less tedious. Also, the text of section 3 is just a little over 100 lines. Three levels of subsubsections seems excessive.

**Reply to comment 5 and 6**: the reviewer has a point that three levels of organization are redudant here. The third level, i.e. the sub-subsections of the evaluation part will be removed in a future version. At the start of section 3, a short paragraph can then be added to discuss the upcoming structure of this section.

**Minor comments**

**Comment 1**: Lines 13–15: "In contrast, the impact of wake losses is exacerbated with increasing capacity density, as the layout-integrated, annual capacity factor varies between 54.4% and 44.3% over the considered range of 3.5 to 10 MW km2." I have read this sentence multiple times, and I am having trouble making sense of it. I think it is phrase "impact of wake losses is exacerbated" that is the problem. Could it be written more clearly?

**Reply**: We suggest an adaptation of this part to: "In contrast, the efficiency losses due to wakes become larger with increasing capacity density..." - this might be less ambiguous.

**Comment 2**: Lines 15–16: "wind farm characteristics and inter-farm distances play an essential role in cluster-scale wake losses, which should be taken into account in future wind farm planning." This sentence is too vague and general to be a meaningful conclusion to your paper. For example, this sentence could be the conclusion of just about every paper in Wind Energy Science. It does not do your study justice.

**Reply**: We agree that a better conclusive statement should be included. We suggest an adaptation of this part to (or a variant of): "In conclusion, our results show that the wake losses in future wind farm clusters are highly sensitive to the inter-farm distances and the capacity densities of the individual wind farms and that the evolution of turbine technology plays a crucial role in offsetting these wake losses.

**Comment 3-6**: *textual improvements*

**Reply**: These valid suggestions will be incorporated in the revised version.

**Comment 7**: Line 71: I think more careful wording of "deep convection is explicitly resolved at the mesogamma scale" is needed. Convective storms may start to be resolved at these scales, but the updrafts comprising that convection are not, as has been shown in Bryan et al. (2003). So, the phrase "deep convection is explicitly resolved" is ambiguous.

**Reply**: This is a valid point. We will be more careful in our wording here: at the mesogamma scale, the model no

longer blocks the explicit development of deep convection due to spatial resolution and excluding the deep convection parametrization at these scales has previously been found not to degrade COSMO simulations even though deep convection is not fully resolved at these scales (Vergara-Temprado et al., 2020).

**Comment 8-18**: *textual improvements*

**Reply**: These valid suggestions will be incorporated in the revised version.

**Comment 19**: Lines 118–120: On what basis can it be said that "the representativeness is high overall, but especially for the wind direction distribution,"? Please evidence that statement.

**Reply**: We will again refer to the supplementary figures mentioned in the previous sentence, these evidence the statement.

**Comment 20-29**: *textual improvements*

**Reply**: These valid suggestions will be incorporated in the revised version.

**Comment 30**: Figure 3: If the top part of the graph is gray (i.e., presumably no difference field plotted there), then why not trim it off? Also, does it make sense to make the graph encompass the same domain as Figure 7? If so, that would be helpful to the readers to compare figures that have the exact same map background.

**Reply**: This would indeed help the comparison, so the map will be adapted.

**Comment 31**: Figure 4: I am having a hard time understanding this graph. There are three colors (light green, dark green, and gray), yet only two colors are explained in the figure caption. Can you write the figure caption more clearly (or replot the graph) to make it easier to understand? I think the histograms are overlapping, but this is difficult to interpret.

**Reply**: It indeed makes sense to adapt the caption like this in the revised version to avoid confusion.

**Comment 32-46**: *textual improvements*

**Reply**: These valid suggestions will be incorporated in the revised version.

---

## Author Comment (AC2)

**Response to the comments of Andrea Hahmann for WES-2023-033**

June 18, 2023

**General feedback from the referee**

The manuscript presents an excellent contribution to assessing wind resources in the North Sea, which could be limited by extracting kinetic energy from the atmosphere by large wind farms offshore. The work is well embedded in the existing literature and brings enough novelty. The design of the study is robust, and results soundly support the conclusions, and I recommend the publication with two 'medium' and a few minor concerns, as follows:

**General reply from the authors**

We thank the referee for taking the time to assess our work and for proposing several adaptations to make the work more robust and clear. In the following we have formulated replies to each of the suggestions after consultation amongst the co-authors.

**Medium comments**

**Comment 1**: It will be nice to get an indication of the accuracy of the simulated model stability classes. Since you are using these frequencies to split the wake losses among stability classes, it would be good to know if they relate to reality. We know the SSTs are input to the model simulation, so could we verify the temperature above the sea using buoy data? For the NEWA project, we estimated that the stability classes could be different by as much as 10% when a different PBL was used in the simulations (see Figures 11-13 in NEWA D4.3 report).

**Reply**: This is a valid concern. As our stability classification relies on vertical temperature profiles, it would be interesting to get a better feel for how accurately COSMO-CLM models these temperature profiles. However, since the stability classification considers data between 50 m and 150 m MSL, a validation over this height range might be more suitable than with buoy data closer to the surface. We propose to add a comparison in terms of the temperature gradient with air temperature measurements at meteomast Ijmuiden for the period 2012-2015 for which we have data at two height levels (21 m and 90 m), hence allowing us to compute a gradient. This can give a first indication of how accurately the temperature profiles are simulated by COSMO-CLM.

**Comment 2**: I am missing a discussion on validating the wake farm parameterisation used. It is challenging to validate these parameterisations in terms of far wakes due to the lack of wind farm data and the fact that current wind farms are not yet as large as the ones you simulate. Please indicate an uncertainty based on the literature. Volker et al. (2015), Fischereit et al. (2022) and other publications show considerable differences between Fitch and EWP schemes with limited validation data, which are relatively close to the wind farm. How much would this uncertainty affect your CF reductions for the North Sea?

**Reply**: We agree that the sensitivity to the wind farm parametrization used deserves more attention. As the mentioned publications point out (in addition to several presentations at WESC2023 around this topic) both the wake development and estimates of wind farm power production are highly sensitive to the parametrization used so we will discuss this uncertainty in our work and include the mentioned publications for this.

**Minor comments**

**Comment 1**: Please follow the WES guidelines for units (e.g., m/s is not acceptable)

**Reply**: Unit formatting will be adapted according the WES guidelines in a revised version.

**Comment 2-3**: textual improvements

**Reply**: We will include these suggestions in the revised version.

**Comment 4**: L94. I would add, "However, wind farms have increasingly affected some of the masts used in the validation. " This is the case for FINO1 and FINO3.

**Reply**: This is indeed only mentioned implicitly in this part in the text. However, on line 164 we go into more detail about filtering of observations for wind farm disturbance and refer to the corresponding supplementary table which summarizes filtered angles or periods per station.

**Comment 5**: Some of your symbols are sometimes italics and sometimes not. e.g. R in 104. Also, after the equations. All symbols should be in italics.

**Reply**: This will definitely be streamlined towards the new version.

**Comment 6**: Is there a direct relationship between the PSS and the EMD? I have a feeling it does. It could be good to mention and thus be able to compare your statistics with those of the NEWA simulations.

**Reply**: We will have a look what the relationship is, but these seem to be applicable to test the same type of agreement indeed. It will be mentioned in the new text.

**Comment 7**: L211: This is not an extrapolation, right? The values above and below the sensor height are known. BTW, this is analogous to a log interpolation of the wind speed between the levels. Too much text confuses people. Also, we want to move away from using power law relationships when data for interpolation is available. The way you write this method could give the wrong impression.

**Reply**: Extrapolation might not be the correct word here, indeed, since information from below and above the sensor height is used. We tested the difference between our method and the log interpolation with a representative dummy example with strong shear: 9.5 m/s at 90 m, 11 m/s at 120 m. The inferred wind speed at 105 m differs with 0.05 m/s between both methods (higher value with our method). Our method also gives 11 m/s at 120 m when inferred from 90 m, so it is a slightly different profile shape going through the two points. In this example, the target level is right in between the two model levels, but in general it will be closer to one of them and we interpolate from the nearest level, further reducing any error. This dummy example is also very strong shear (e.g. compare with average profiles Fig. 6), so in general the differences between the methods will be smaller. To conclude, the differences between both methods introduces some additional uncertainty but since differences are very small, it is expected that it does not impact the findings of our study. It is probably also difficult to determine which method is better for our use case given the other uncertainties on measurement data.

**Comment 8-9**: Textual improvements.

**Reply**: We will include these suggestions in the revised version.

**Comment 10**: How are the turbines located in each grid cell? That should be mentioned.

**Reply**: This will be added to the text. In the used implementation of the Fitch WFP, no subgrid-scale layout effects are included. All turbine rotors are assumed perpendicular to the wind and there are no subgrid-scale turbine wake interactions.

**Comment 11**: Extrapolation of each measurement has been shown to work poorly. See Badger et al. (2016) in JAMC, DOI:10.1175/JAMC-D-15-0197.1

**Reply**: We assume that this comment refers to the vertical extrapolation of in situ data to the 10 meters MSL of the ASCAT data. It is true that this constant coefficient extrapolation is too simplistic as pointed out by the referenced publication, so that stratification effects are not taken into account. A mention of this limitation will be added to the text.

**Comment 12**: L258-259: Please be more explicit on what data is used to compute the static stability

**Reply**: We will add additional information on the number of height levels in the considered range (see line 245), about the temporal resolution and more detailed information on the grid cell used to select timesteps in the observational dataset in the revised version.

**Comment 13**: Figure 2. Please explicitly name the period used.

**Reply**: We will make more explicit that it refers to the measurement period of each individual measurement station. Adding the individual time periods to the figure might become messy? Instead we have added this information to Appendix A1 and A2 and in visual form to Fig. S4.

**Comment 14**: Figure 7. The height at which the maps are computed and the time period details are also missing.

**Reply**: This will be made more explicit towards the next version.

**Comment 15**: L360-361. It is not clear what you mean by "data points". Are these in space or in time? I guess in time, but maybe a time match will be best or a frequency.

**Reply**: This is indeed not very clear. It refers to time points, but adopting a frequency instead will be better. We will adapt this.

**Comment 16**: I like the transects in Fig. 8. Especially because they help assess the necessary spacing between wind farm clusters regarding recovery distances. But again, you are missing the height in Fig 8 caption. Please make sure that the captions are complete.

**Reply**: It is the hub-height of the considered scenario, but should indeed be made more explicit for the reader to avoid confusion.

---

## Author Response (AR1)

**Revision updates for WES-2023-33**

September 15, 2023

**Note to the editor**

Since the initial submission of the manuscript, some additional points of improvement were identified next to the suggestions of the referees. Most of these points are minor, however, as already discussed with the associate editor, one major point of improvement was identified. We thank the associate editor for the patience and the supportive attitude in dealing with this major point. Therefore, this document contains three sections: first the updates are described that were done and were not related to the referee comments (section 1). This is then followed by a list of comments and replies to the first referee (section 2) and then finally the same for all the comments of the second referee (section 3).

**1. Additional updates after the initial submission**

**Point 1:** A code bug led to the modelled thrust being too low in the WFP implementation compared to the initial description of Fitch (2012).

**Update**: Testing indicated significant differences with the initial simulations. Therefore, the four wind farm simulations were done again with the code bug fixed. As a consequence, the results that were mentioned in the abstract, results and discussion and conclusion sections have been adapted. The general change is that wind farm wake effects are now stronger and all signals have been amplified to some extent. The text body, Figures 7 (bottom panel), 8, 9 and 10 and Table 4 have now been adapted with the new data. Most conclusions have remained the same, albeit with stronger magnitudes. However, in some locations the text body, sentences no longer valid have been removed and have been replaced. This is visualized in detail in the provided latexdiff document (mainly section 3.2). At the bottom of the manuscript, a code availability statement was added, referring to the code and the implementation guide (which includes an update log of all the updates compared to the previous version). The new WFP implementation in COSMO5.0 has been added as a next version to the repository of the intial code: https://www.wdc-climate.de/ui/entry?acronym=WindFarmPCOSMO5.0clm15

**Point 2:** Simulation and evaluation domain depicted incorrectly in Fig. 1 due to inadequate .csv description of the domain.

**Update**: A better domain description was generated and the outlines of the domains have been adapted accordingly in Fig. 1.

**Point 3:** The amount of grid cells that were clipped at the edges to remove the relaxation and spin-up zones was too low.

**Update**: An extra 10 grid cells were now clipped at the edges. This has slightly changed the results of Fig. 7 and table 4. For Table 4, this mainly pertains to columns 3 and 6. In column 3, the total capacity in the evaluation domain is slightly reduced and in column 6, the layout-integrated AEP has reduced accordingly. Importantly, results in column 6 are also influenced by the changes related to point 1 of this document. The impact on all other results, which are mainly efficiency indicators, is negligible.

**Point 4:** The computation of the the capacity factors for the full domain per grid cell (only Fig. 7 and table 4) had a small error in the python code leading to slightly too low capacity factors.

**Update**: The code was fixed leading to slightly higher capacity factors (a few percents). It was checked that indeed this error was not present in the computation of all other results.

**Point 5:** Acknowledgements needed to be added for co-author Naveed Akhtar.

**Update**: One acknowledgement has now been added.

**2. Comments and updates for referee 1: David Schultz**

**General feedback from the referee**

This is an important paper because it shows that the increase in height of the turbines from 5 MW to 15 MW more than offsets the loss due to wakes. I approach this review as a meteorologist and mesoscale modeler. Hence, my comments are mostly focused on these aspects of the manuscript. I see no problem with the science and the results, only the presentation.

**General reply**

The authors thank the referee for the thorough assessment of the text and the suggested adaptations to improve the presentation of our results and the text body. We believe that after having processed these comments, the presentation and text are now clearer and more streamlined across the manuscript which is very much appreciated. We summarize here all the updates done to the initial submission based on the different comments of the reviewer.

**Major comments**

**Comment 1**: The conclusion section is unbalanced. Lines 414–423 represent an unacceptably short, incomplete, and qualitative summary of the first part of the study. In contrast, lines 424–443 represent a much more detailed and quantitative set of conclusions. I prefer the latter, as I imagine most readers would who would want to read the conclusion to get a more clear picture of the results of this study. I suggest a revision to the first part of the conclusion section.

**Update**: The evaluation part of the conclusion section has now been elaborated: more details are provided with a focus on more quantitative information. This has made the conclusion more balanced between the two parts of the study and we thank the reviewer for pointing this out. The conclusion of the evaluation part is now:

*First, the model skill in simulating the wind climate was evaluated in a comparison with in situ, lidar and satellite data, which revealed that*

- *The differences between the measured and modelled, long-term mean wind speed at turbine hub height ($\sim$100 m) are generally within the measurement uncertainty. This is also the case for differences at higher altitudes (up to 290 m), but closer to the surface COSMO-CLM underestimates the mean wind speed ($\sim$ -0.5 ms$^{-1}$). Under stable stratification ($\sim$25%), the model underestimates the long term mean wind speed at turbine height, but not under weakly stable and unstable stratification ($\sim$75%).*

- *The agreement between the measured and modelled, long-term wind speed histograms is high, with a PSS above 95% in most cases. The theoretical capacity factors derived from these histograms agree well overall, but small underestimations ($\sim$1–5%) are present at some locations.*

- *The agreement with the wind speed measurements is consistent over the different years of the simulation period as inter-annual variations in the mean wind speed difference and the PSS are small. Also the seasonal variability in the shape and location of the wind speed distribution is captured by COSMO-CLM.*

- *Multi-year histograms of wind direction also agree well, with again a PSS above 95% in most cases. Also the variation of the wind speed histograms over 12 directional bins (30°) are adequately captured in the model. This encourages the application of COSMO-CLM to wind farm modelling as wind farm shapes are adapted to the regional wind climate.*

*As deviations mainly occur under stable conditions, a stability-dependent bias correction could be considered for future applications in addition to continuous efforts to improve the model. Overall, this evaluation emphasizes the value of having a large set of wind measurements available in regions for offshore wind farm development, as it allows a benchmarking of mesoscale models over the region of interest.*

**Comment 2**: Figure 2: The color scale needs work. First, the zero point should be white, not bluish-yellow, to indicate its true neutrality. It's hard to interpret otherwise. Second, the color scheme is not symmetric. Negative

values are all shades of blue, but positive values are yellow, orange, and red. Instead, all positive values should be shades of red, opposite of the negative values. In any case, I recommend to the authors to avoid the rainbow color scheme. It distorts gradients, among other issues. Please fix similar problems with Figures 3 and S5.

**Update**: The color scale of Fig. 2, Fig. 3 and Fig. S5 (now Fig. S6 in the revision) has been replaced with a blue-white-red scale, which is more symmetric and has white as the zero point. While still considering colorblindness, this color scale now makes the figures easier to interpret. The new figures are included in the revised manuscript.

**Comment 3**: An excessive number of grid lines appears on Figures 5, 6, S1–S4, and S6–S9. These are distracting from the data (i.e., what Edward Tufte calls chartjunk) and should be eliminated.

**Update**: The number of gridlines has been minimized for Fig. 5, Fig. 6, Fig. S1, S2, S3 and S6 (now Fig. S7 in the revision). For Fig. S4, S7-S9, the gridlines have been kept the same since we believe it is easier to interpret the figures with these present. The figures are included in the revised manuscript.

**Comment 4**: Model output and observations have different colors in each figure where they are compared directly against each other: Figures 4, 5, and 6 (and comparable figures in the supplement). Can a similar color scheme (red for model and black for observations, for example) be employed across all three of these figures? It would sure simplify things for the reader having that consistency across all the figures.

**Update**: Figures 4, 5 and the bottom part of figure 6 have now all the combination: grey for the observations - orange for the model. This was found to be the best choice for streamlining the color combination across the relevant figures, also considering colorblindness. The figures are included in the revised manuscript.

**Comment 5**: Lines 282–284: I find three levels of organization (section 3 to subsection 3.1 to subsection 3.1.1) without any text introducing each confusing. The authors need to put these sections into context before diving down three levels to a very specific quantity (e.g., wind speed at 290 m). For example, after the title for section 3, there should at least be a paragraph explaining how this section is structured and any general information that pertains to it. Also, after the title for section 3.1, there should at least be a paragraph explaining what will be discussed in this subsection and any general information that pertains to it. The same is true for section 3.2. The text just jumps right in with results from Figure 7. Can you provide some context to the reader first?

**Update**: At the start of the two sub-sections a short, situating description has been added for the reader. For the evaluation section 3.1:

*This subsection covers the model performance evaluation. First, the general evaluation based on all validation sources and the complete height range (10 m to 290 m) is described. This is followed by a more detailed performance analysis at turbine hub-height (∼100 m) and finally the evaluation is extended to the different atmospheric stability classes.*

And for the wind farm simulation intercomparison, i.e. section 3.2:

*This subsection covers the results of the wind farm simulations. First, the impact of the NREL8.1 base scenario on the wind climate and wind resource is described, also under different atmospheric stability conditions. Afterwards, the different wind farm scenarios are compared in terms of cluster-scale wake effects and efficiency of power production.*

**Comment 6**: Line 284: Are three levels of organization necessary? Usually sections and subsections are sufficient. Could you just have different sections for results? Section 3.1 becomes section 3, section 3.2 becomes section 4, etc.? Alternatively, you could just drop the third level of subsubsections. I'm not sure they are helping the reader. It would make the text more readable and less tedious. Also, the text of section 3 is just a little over 100 lines. Three levels of subsubsections seems excessive.

**Update**: The sub-subsection organization of the evaluation part, i.e. 3.1.1, 3.1.2 and 3.1.3 have now been removed so that all these parts are merged to sub-section "3.1 evaluation", hence only two levels of organization remain here.

**Minor comments**

**Comment 1**: Lines 13–15: "In contrast, the impact of wake losses is exacerbated with increasing capacity density, as the layout-integrated, annual capacity factor varies between 54.4% and 44.3% over the considered range of 3.5 to 10 MW km2." I have read this sentence multiple times, and I am having trouble making sense of it. I think it is phrase "impact of wake losses is exacerbated" that is the problem. Could it be written more clearly?

**Update**: This has been adapted to: "In contrast, the efficiency losses due to wakes become larger with increasing capacity density as the layout-integrated, annual capacity factor varies between 51.8% and 38.2% over the considered range of 3.5 to 10 MW km$^{-2}$"

*In contrast, the efficiency losses due to wakes become larger with increasing capacity density as the layout-integrated, annual capacity factor varies between 51.8% and 38.2% over the considered range of 3.5 to 10 MW km$^{-2}$"* - this is indeed clearer.

**Comment 2**: Lines 15–16: "wind farm characteristics and inter-farm distances play an essential role in cluster-scale wake losses, which should be taken into account in future wind farm planning." This sentence is too vague and general to be a meaningful conclusion to your paper. For example, this sentence could be the conclusion of just about every paper in Wind Energy Science. It does not do your study justice.

**Update**: We improved the conclusive statement to:

*In conclusion, our results show that the wake losses in future wind farm clusters are highly sensitive to the inter-farm distances and the capacity densities of the individual wind farms and that the evolution of turbine technology plays a crucial role in offsetting these wake losses.*

**Comment 3**: Lines 25, 33: Why these three citations of all the citations that have been published on the efficiency of wind-farm wakes? In such instances, it would help to precede the list of citations with "e.g." to indicate that these are a sampling of all the possible sources that could have been cited. You may wish to consider adding "e.g." to other lists of citations, as well.

**Update**: An "e.g." has been added to all the citations in the text where this comment is relevant

**Comment 4**: Line 64: In contrast, the "e.g." is not needed here because presumably there is only source for "documentation".

**Update**: The "e.g." is retained because it refers to only one of the chapters of the model documentation.

**Comment 5**: Line 47: should be "and/or", with no spaces

**Update**: Changed as suggested.

**Comment 6**: Line 67: The verb tense changes back and forth from present to past tense in this paragraph. I think past tense sounds better, but whatever you pick, aim to be consistent.

**Update**: This has been adapted to the past tense throughout the paragraph except where another tense (e.g. present) is really needed.

**Comment 7**: Line 71: I think more careful wording of "deep convection is explicitly resolved at the mesogamma scale" is needed. Convective storms may start to be resolved at these scales, but the updrafts comprising that convection are not, as has been shown in Bryan et al. (2003). So, the phrase "deep convection is explicitly resolved" is ambiguous.

**Update**: This has been adapted to: "At the meso-$\gamma$ scale, the model resolution partly allows the explicit development of deep convection so that only shallow convection was parametrized according to the scheme of Tiedtke (citation). Switching of the deep convection parametrization on this resolution has previously been shown not to degrade COSMO simulations (Vergara-Temprado et al., 2020).

**Comment 8**: Lines 80, 340: "Hence" cannot be used as a conjunction in this context.

**Update**: Updated for both cases.

**Comment 9**: Line 89: I'm unclear what role these two citations are supporting. Period of 13 years? Need for sampling inter-annual variability? Other studies doing model evaluation? More explanation is needed for why their citation is relevant to this sentence. That may involve rewriting the sentence so that the reason for the citations becomes more clear.

**Update**: This sentence has been rewritten to make this more clear. The adapted version is: "To evaluate the model performance, a simulation was performed for a period of 13 years (2008–2020). Data from in situ, lidar and satellite measurements over the North Sea are abundant in both space and time for this period. Additionally, the length of the simulation ensures that the inter-annual variability in the wind conditions, which has been widely described (e.g. Geyer et al., 2015; Ronda et al., 2017), is sampled well."

**Comment 10**: Lines 93, 264, 367, 396: "Cf." means "compare". So what is being compared to section 2.4.1 It isn't clear. Write instead "cf. A and B" to compare A to B.

**Update**: We recognized that "cf." was used incorrectly in our text. All instances in the text have been changed to not include this.

**Comment 11**: Line 99: Change "constructions" to "construction".

**Update**: Adapted accordingly.

**Comment 12**: Line 104: Delete the comma.

**Update**: Adapted accordingly.

**Comment 13**: Lines 107, 112, 213, 216, 243, 269, 402, 424: Please insert noindent before "where".

**Update**: All instances in the text have been adapted with noindent.

**Comment 14**: Lines 104 and elsewhere: Italicize R, as it is a variable

**Update**: All variables in the manuscript, such as $R$ but also $PSS$ have now been italicized in the text body.

**Comment 15**: Line 112: Italicize b.

**Update**: Has now been italicized.

**Comment 16**: Lines 109, 116, and elsewhere: Change hyphens to en dashes (i.e., two hyphens in LaTeX) to connect two items in a range. Fix also in Tables A1 and A2. Fix throughout the manuscript.

**Update**: All instances throughout the text where this update was needed have been adapted to en dashes.

**Comment 17**: Line 118: Capitalize "Figures"

**Update**: This has been capitalized now.

**Comment 18**: Lines 118–120: I am having a hard time understanding this sentence. I think there is too much being communicated within. Break it up, perhaps.

**Update**: This sentence has now been broken up as suggested. The new version is: "Based on this procedure, the year 2016 was selected for the simulations, as the representativeness is high overall for this year (supplementary Fig. S1). In addition, the representativeness is especially high for wind direction (supplementary Fig. S2), which is particularly important for the study of inter-farm wake interactions."

**Comment 19**: Lines 118–120: On what basis can it be said that "the representativeness is high overall, but especially for the wind direction distribution,"? Please evidence that statement.

**Update**: References to supplementary figures have now been added to these statements. These were previously only referred to in the previous sentence, but have now been included explicitly for these statements.

**Comment 20**: Lines 157, 219, 230: Is this a new paragraph? If so, indent it. If not, then combine it with the previous line.

**Update**: For each instance this has now been evaluated and corrected if indeed it was still belonging to the previous paragraph.

**Comment 21**: Line 164: Change "since" to "because" to avoid implication of time that "since" implies. See also lines 207, 239, 257, 266.

**Update**: This has been adapted to "because" for all the instances in the text where the implication of time was indeed wrong.

**Comment 22**: Line 170: spell out "including"

**Update**: This has been done as proposed.

**Comment 23**: Lines 195, 311, 445, 447: Change "while" to "although" to avoid the implication of simultaneity that "while" implies.

**Update**: This has been adapted to "although" for all the instances in the text where the implication of time was indeed wrong.

**Comment 24**: Line 199: Change "resolution" to "grid spacing" for consistency and precision

**Update**: This has been done as proposed.

**Comment 25**: Line 208 and throughout the manuscript: Hyphenate "10-minute period". Fix throughout.

**Update**: This has been hyphenated for all instances in the text.

**Comment 26**: Lines 214, 241, 267: Delete the colon

**Update**: These have been removed.

**Comment 27**: Line 270: Italicize P.

**Update**: This has been done.

**Comment 28**: Line 295: "further" should be "farther".

**Update**: This has been adapted.

**Comment 29**: Line 297–299: This sentence should cite Fig. 3. You were talking about Figure 3, but then cited Figure 2. You should return to citing Figure 3 to avoid any confusion and point the reader to the right figure.

**Update**: The figure referencing here was indeed not clear enough. The references per sentence have now been updated to fix this, now including the correct use of "cf." ;). The new version is: "The model underestimation near the surface that was identified against the in situ data in the southern North Sea is much smaller than the differences compared to ASCAT in this region (cf. Fig. 2 and Fig. 3). A three-way comparison with three in situ stations shows that the mean differences against the in situ data exceed the in situ measurement uncertainty for both COSMO-CLM and ASCAT (Fig. 3)."

**Comment 30**: Figure 3: If the top part of the graph is gray (i.e., presumably no difference field plotted there), then why not trim it off? Also, does it make sense to make the graph encompass the same domain as Figure 7? If so, that would be helpful to the readers to compare figures that have the exact same map background.

**Update**: This map has been adapted in such a way as to balance the two suggestions of the referee: The northern gray zone has been clipped, but at the same time matched to the maps of Fig. 7 i.e. the actual extent of the evaluation domain.

**Comment 31**: Figure 4: I am having a hard time understanding this graph. There are three colors (light green, dark green, and gray), yet only two colors are explained in the figure caption. Can you write the figure caption more clearly

(or replot the graph) to make it easier to understand? I think the histograms are overlapping, but this is difficult to interpret.

**Update**: The caption of this figure has been updated to explain the three different colors visible in this plot.

**Comment 32**: Line 315: Is "stable" the right word? "Consistent" would be better.

**Update**: This has indeed been updated to "consistent".

**Comment 33**: Line 343: The title of this subsection is vague: "characteristics". It could be more clear what characteristics are being varied.

**Update**: This title has been updated from "Effect of wind farm characteristics" to: "Impact of wind farm characteristics on cluster-scale wake losses". The title is now more concrete and less vague, however the characteristics have not been written out explicitly because, as described in the methods section, many factors are investigated: turbine type, capacity density, inter-farm spacing and wind farm size. The title would be too long if we include all of these, but we believe that the abstract, introduction, methods, conclusion and the results section itself make clear enough what characteristics we investigate.

**Comment 34**: Line 347: Put a comma between "conditions capacity".

**Update**: This has been added.

**Comment 35**: Line 360: Change "while" to "whereas" to avoid implication of simultaneity.

**Update**: Changed accordingly.

**Comment 36**: Line 364: Why is "weakly" in parentheses? You don't include the parentheses when that term is used in line 366. I suggest deleting the parentheses.

**Update**: This was to refer to both the weakly stable and stable classes simultaneously, but was actually a poor choice. This has now been adapted to "stable and weakly stable".

**Comment 37**: Figure 8 caption: Use the $\pm$ symbol.

**Update**: This has now been used instead.

**Comment 38**: Lines 373, 384: Delete "(blue)" as that information belongs in the legend and figure caption, not the text.

**Update**: This has been removed.

**Comment 39**: Line 387: Change "wit" to "with".

**Update**: This has been corrected.

**Comment 40**: Line 404: Be careful of the difference between cite and citep.

**Update**: These citations have been corrected.

**Comment 41**: Line 404: Insert a comma after "IEA8.1".

**Update**: comma has been inserted.

**Comment 42**: Line 405: Insert a comma after "IEA10.0".

**Update**: comma has been inserted.

**Comment 43**: Lines 414–415: "negative bias" in "wind climate" is unclear. Can you just say that the "model was underpredicting the wind speed"? That is easier to understand. Look for other similar opportunities throughout the manuscript to convey your message more simply and more clearly.

**Update**: This piece of text has been updated and, as suggested, we have gone through the evaluation parts of the study to convey the message in a more easily understandable way. This is noticeable in the latexdiff document.

**Comment 44**: Line 417: A comma is needed after "95%" to join the two independent clauses

**Update**: This comma has been added

**Comment 45**: Line 417: Delete "both" because you describe three things: "seasonal, inter-annual, and directional variability".

**Update**: This has indeed been removed.

**Comment 46**: Lines 418–419: This sentence is unclear. Also, what about biases in other stability conditions?

**Update**: The sentence "Separation into stability classes reveals small, height-dependent biases under dynamically stable conditions" has been updated within the updated conclusion to: "... Under stable stratification ($\sim$25%), the model underestimates the long term mean wind speed at turbine height, but not under weakly stable and unstable stratification ($\sim$75%)."

**3. Comments and updates for referee 2: Andrea Hahmann**

**General feedback from the referee**

The manuscript presents an excellent contribution to assessing wind resources in the North Sea, which could be limited by extracting kinetic energy from the atmosphere by large wind farms offshore. The work is well embedded in the existing literature and brings enough novelty. The design of the study is robust, and results soundly support the

conclusions, and I recommend the publication with two 'medium' and a few minor concerns, as follows:

**General reply from the authors**

The authors thank the referee again for taking the time to go through our work and formulate some very insightful suggestions to help us in making our work more robust and clear. Here we summarize the different updates that have been done to the initial submission based on the different comments of the referee.

**Medium comments**

**Comment 1**: It will be nice to get an indication of the accuracy of the simulated model stability classes. Since you are using these frequencies to split the wake losses among stability classes, it would be good to know if they relate to reality. We know the SSTs are input to the model simulation, so could we verify the temperature above the sea using buoy data? For the NEWA project, we estimated that the stability classes could be different by as much as 10% when a different PBL was used in the simulations (see Figures 11-13 in NEWA D4.3 report).

**Update**: As we proposed in the reply, we have handled this concern by performing a temperature gradient validation at Meteomast Ijmuiden. We believe that a validation over the available height range (21m-90m) is more suitable than close to the surface, because it is closer to the height range over which our stability criterion is evaluated. This validation was done for a 4-year period period (2012–2015). An additional supplementary figure (supplementary Fig. S5) has now been added which shows the long-term histograms of the temperature gradient for the measurements at the station and the corresponding model data. Although the match is not perfect, the agreement is good (along with a negligible difference in the mean) which supports the validity of doing the stability subdivision with the Bulk Richardson criterion as we did it. At line 257 in the old manuscript the following text has been added to discuss this comparison:

*A comparison of the modelled temperature gradients with measured temperature gradients at the station MMIJ between 90 m MSL and 21 m MSL shows a good correspondence in the long-term temperature gradient probability distribution, indicating sufficient model skill for this subdivision into stability classes (supplementary Fig. S5).*

And the added figure is:

[Figure]

Figure 1: Histograms of the instantaneous temperature gradient between 90 m MSL and 21 m MSL based on 10-minute data for the period 2012-2015 at the location of measurement mast Ijmuiden (MMIJ). Lightgreen: only COSMO-CLM; Grey: only the measurements; Green: overlap between the histograms.

In addition to this, another sentence is added right after this which mentions the good temporal correlation of modelled and measured temperature gradients (Pearson correlation coefficient of 0.85) to warrant our choice to use the same timesteps for each stability class in the measurements as were determined based on the stability computations for the model:

*Because vertical profiles of pressure and temperature are generally not available over the range of the meteorological masts or wind lidar scanning ranges, the stability criterion can only be computed for the model. Based on a good temporal correlation between the temperature gradients of COSMO-CLM and measurement mast MMIJ (Pearson correlation coefficient = 0.85), the timesteps matched to a stability class for the model grid cell nearest to each measurement location were also matched to that stability class for the measurement data.*

**Comment 2**: I am missing a discussion on validating the wake farm parameterisation used. It is challenging to validate these parameterisations in terms of far wakes due to the lack of wind farm data and the fact that current wind farms are not yet as large as the ones you simulate. Please indicate an uncertainty based on the literature. Volker et al. (2015), Fischereit et al. (2022) and other publications show considerable differences between Fitch and EWP schemes with limited validation data, which are relatively close to the wind farm. How much would this uncertainty affect your CF reductions for the North Sea?

**Update**: This very important aspect deserved more attention in our text. Based on an assessment of literature, we find indeed a significant variation across WFP schemes, but also that the evaluations indicate a very good performance of Fitch in terms of modelled wind speed deficits in and behind wind farms, often outperforming other WFP schemes. Two sections were added to the text body which discusses this based on literature. First, in the methods section:

*Several other wind farm parametrizations exist (Fischereit et al., 2022) and it has been shown that the modelled wind speed deficits inside and behind a wind farm can vary substantially from the Fitch WFP (Ali et al., 2023). However, validation of the Fitch WFP with offshore masts, lidars and airborne measurements in the wake of a wind farm has shown very good performance for HARMONIE-AROME as wind speed biases are strongly reduced (Van Stratum et al., 2022; Dirksen et al., 2022). This good performance has also been determined in WRF by comparing to offshore masts*

*(Garcia-Santiago et al., 2022) and in COSMO-CLM by comparing to LES (Chatterjee et al., 2016) and airborne measurements (Akhtar et al., 2021). Also wind speed reductions inside of a wind farm have been shown to agree well with airborne measurements (Ali et al., 2023), mast measurements (Dirksen et al., 2022) and RANS simulations (Fischereit et al., 2021). Moreover, comparisons with other WFP schemes show that Fitch generally outperforms these other schemes, both inside a wind farm and in the farm wake (Fischereit et al., 2021; Ali et al., 2023).*
And also in the conclusion section, this point is adressed again with the following part:

*Whereas comparisons between wind farm parametrizations have shown large variations in terms of modelled wind speed deficits inside and behind wind farms (Ali et al., 2023), validation efforts in several mesoscale models have indicated a very good performance of the Fitch WFP (Fischereit et al., 2021; Van Stratum et al., 2022; Ali et al., 2023). Nonetheless, the use of other WFP schemes might significantly alter the magnitudes presented here, more so due to the large clusters and large wind farms included in the considered layout which can even lead to wake losses for background wind speeds well above rated. Hence, further benchmarking studies of WFP's for a range of atmospheric conditions and validation data could help in further reducing this WFP-related uncertainty. An additional complication here is that this study includes wind farms of non-existent sizes for which validations simply do not exist.*

**Minor comments**

**Comment 1**: Please follow the WES guidelines for units (e.g., m/s is not acceptable)

**Update**: All the units have now been adapted according to the guidelines.

**Comment 2**: L22: change are to is

**Update**: After inspection, "are" still seems like the adequate option here since there are two subjects, so it was retained.

**Comment 3**: L28: "and gigawatt-scale wind farms emerge..." I find that maybe the verb tense is not right, future?

**Update**: This sentence has been adapted to be more clear in general, thereby also improving the tense use. The updated version is: "Currently, limited space and the urgent decarbonization of electricity systems lead to the installation and planning of very dense wind farms (capacity density $> 10$ MW km$^{-2}$) and exceptionally large wind farms (capacity $> 1$ GW) that are strongly impacted by these turbine interactions (citations)."

**Comment 4**: L94. I would add, "However, wind farms have increasingly affected some of the masts used in the validation. " This is the case for FINO1 and FINO3.

**Update**: This is indeed only mentioned shortly in this part in the text. However, in section 2.4.1 we go into more detail about filtering of observations for wind farm disturbance and refer to the corresponding supplementary table which summarizes filtered angles or periods per station. So, we decided not to add another mention of this here.

**Comment 5**: Some of your symbols are sometimes italics and sometimes not. e.g. R in 104. Also, after the equations. All symbols should be in italics.

**Update**: This has now been streamlined across the complete text: all variables are now consistently italicized in the text body.

**Comment 6**: Is there a direct relationship between the PSS and the EMD? I have a feeling it does. It could be good to mention and thus be able to compare your statistics with those of the NEWA simulations.

**Update**: The relationship between the PSS and the EMD metric has now been added to the text to support comparability between these metrics. The following mention was added to line 113 of the initial submission: "... For one-dimensional histograms, this metric is connected to the Earth Mover's Distance (EMD) metric, which in contrast represents the area of mismatch between two histograms (Rabin et al., 2008)"

**Comment 7**: L211: This is not an extrapolation, right? The values above and below the sensor height are known. BTW, this is analogous to a log interpolation of the wind speed between the levels. Too much text confuses people. Also, we want to move away from using power law relationships when data for interpolation is available. The way you write this method could give the wrong impression.

**Update**: The word "extrapolation" was indeed updated to "interpolation". As discussed in the reply to the referee, a test has indicated negligible differences between both interpolation methods so that there are no implications for the findings of our study. Adding a discussion on this topic in the text body seems out of the scope of this manuscript, definitely since the reply to the referee on this topic is publicly available.

**Comment 8**: L241. Please use conventional abbreviations (for example, from textbooks) for often-used quantities, e.g. $Ri_B$.

**Update**: Adapted for the Bulk Richardson number throughout the text based on the textbook of Stull (1988) to $R_B$.

**Comment 9**: Captions of Fig 8-10: Grey shadings represent wind farm locations.

**Update**: This has been updated for all relevant captions

**Comment 10**: How are the turbines located in each grid cell? That should be mentioned.

**Update**: To include this information, a sentence was added after line 105 of the initial submission: "Because the wind farm parametrization assumes that turbines within a single grid cell never have any wake interactions, no additional information is required on the layout of the turbines in each wind farm."

**Comment 11**: Extrapolation of each measurement has been shown to work poorly. See Badger et al. (2016) in JAMC, DOI:10.1175/JAMC-D-15-0197.1

**Update**: It is true that this constant coefficient extrapolation is too simplistic as pointed out by the referenced publication, because stratification effects are not taken into account. This is a limitation of this part of the evaluation component in our study, but we have tried to mitigate this by selecting the stations with a measurement height below 30 meters height, so close to the target height of 10 m, in order to reduce the impact of extrapolation errors. While there are stations in our study with even lower measurement heights (i.e. < 20 m), these are located too close to the coast, so that no ASCAT data is available in these locations.

**Comment 12**: L258-259: Please be more explicit on what data is used to compute the static stability

**Update**: Additional details have been provided on the amount of model levels in the considered height range. Also more details were provided on the stability subdivision of the measurement data.

**Comment 13**: Figure 2. Please explicitly name the period used.

**Update**: The description in the caption was adapted to make more clear that the data is for the full measurement period of each individual station and a reference to the tables holding this period information was included in the caption. Adding the individual time periods to the figure would also make the figure more messy, we think.

**Comment 14**: Figure 7. The height at which the maps are computed and the time period details are also missing.

**Update**: The heights have been explicitly stated now for all relevant figures (also supplementary).

**Comment 15**: L360-361. It is not clear what you mean by "data points". Are these in space or in time? I guess in time, but maybe a time match will be best or a frequency.

**Update**: This concerned time points. For more clarity, frequencies were now adopted as suggested by the referee.

**Comment 16**: I like the transects in Fig. 8. Especially because they help assess the necessary spacing between wind farm clusters regarding recovery distances. But again, you are missing the height in Fig 8 caption. Please make sure that the captions are complete.

**Update**: For figures 8 and 9 and Table 4 these heights have now been added explicitly in the captions.

---

## Author Response (AR2)

**Response to the second round of comments for WES-2023-33**

December 14, 2023

**general reply**

We want to thank the two referees and the editor for reviewing our revised submission. We are pleased to see that our changes are appreciated and we agree that this has made the work a lot stronger. Here we address the five additional comments that were provided by the referees and the editor.

**replies to comments**

**Comment 1**: Validation is done against wind speed but nothing about wind direction. In order to provide a complete understanding into the validity of the models, the authors can reproduce something similar to Figure 6 that looks into wind direction.

**Reply**: It is true that the validation part of our research has a strong focus on the variable wind speed. However, we do supply a validation for wind direction as well in our manuscript. The results of this validation are summarized in Table 3 of the manuscript, where the mean bias is assessed, next to the match in the wind direction distributions spanning the entire measurement period for each station. In addition, there is also a validation of the wind-direction-dependent wind speed distributions (Fig. S7), which also gives an indication of how well the directional variability is modelled by COSMO-CLM. Nonetheless, an equivalent of Fig. 6 for wind direction is a very interesting suggestion, but since our wind direction validation already covers measurement heights from 62 m to 120 m, the analysis is already implicitly multi-height. A sentence was added to the manuscript at line L340 to highlight this

*Because the considered measurements vary substantially in measurement height, i.e. from 62 m MSL up to 120 m MSL, this comparison indicates consistency of the good performance with height.*

We have investigated the possibility to clarify the wind direction validation part of the research more, but in the previous round of revisions we already added a subsection in the conclusion which summarizes the model performance in terms of wind direction (L441-L443) as it was asked to expand here more on our validation. Also in the abstract the model's performance for wind direction is briefly mentioned (L6-L8).

**Comment 2**: Results from figure 8 are somehow counter-intuitive. One would expect that during unstable conditions wakes recover notably faster than during stable conditions. However, we see that this does not happen in the results from TR4 and some parts of TR3. Can authors comment how or why in TR4 the onset velocity close to 0 km is lower for the stable case which then leads to lower velocity deficits (100-150km)? Similarly to TR3, where stable and unstable conditions seem to feature similar wake recovery rate.

**Reply**: This is a valid and interesting comment, which has also popped up in our discussions. We have done several sensitivity tests to how the transects were computed, but the impacts were always small. The lower onset speed in TR4 is only ∼0.2 m/s and could be influenced by the presence of the wind farm to the south of the southwestern tip of TR4, which is not present in the NOWF simulation (Recall that there is a tolerance angle of 15° around the transect orientation for the inclusion of samples). Similarly, small deviations at the transect tips can be related to farm wake advection / lateral speed-up effects related to the wind farm presence.
It is true that the recovery differences between stability classes are not always according to what is typically found in literature. However, our analysis differs from studies focusing on stability-dependent wake recovery in several aspects. First of all, we look at time-averages of tens to hundreds of samples and not at specific cases. Secondly, the collection of samples in each stability class is a collection of different wind speeds. Thirdly, we consider a very large transect (multi-100km) where the stability criterion and wind direction criteria is evaluated only at the center of each transect so that along-transect variations in wind direction and stability can occur. And finally, we consider wind farms of non-existent sizes, further hampering the

comparison with studies based on present-day wind farms.

Nonetheless, we do clearly see that over the wind farms, the mean wind speed reduction is generally weaker under unstable conditions, which does point to enhanced mixing of momentum under unstable conditions. More so because the wind farm forcing actually leads to the strongest relative wind speed reductions for unstable conditions according to the following reasoning: The term $V * C_t$ in the wind farm parametrization, which determines the relative wind speed reduction (Fitch et al., 2012), is largest for the rated wind speed and the wind speeds under unstable conditions are most similar to the rated wind speed of the three stability classes. According to our quantifications the mean relative reductions should be 5% to 20% stronger for unstable conditions along transects TR1, TR2 and TR4 because of this.

Furthermore, the large wind farms considered in this simulation also add considerable TKE to the boundary layer and in relative terms more so for stable conditions because it is for these sub-rated wind speeds that $Ct - Cp$ is largest. The advection of this large TKE amount behind the wind farm might further enhance the wake recovery under stable conditions. Also, since the stability criterion is computed for the NOWF simulation, this analysis does not consider the modification of the atmospheric boundary layer (incl. stability) due to the wind farm forcing, whereas it is possible that this forcing leads to e.g. strongly enhanced mechanical turbulence also under stable conditions.

To conclude, our stability-based analysis of the wind farm runs show that the annual impacts of the wind farm forcing are substantial for all stability classes, but that unstable conditions show lower deficits over the wind farm and by extension often also lead to smaller reductions at the inflow of the next wind farm. It does not clearly show differences in wake recovery, but there are substantial differences with how this is analyzed in other studies and other research setups would be more suitable to investigate the topic of stability-dependent wake recovery. To address this topic in the manuscript, the following text was added in the results and discussion section, after line L380

*However, the transects do not show a significantly slower wind farm wake recovery for stable conditions, as has been found based on observations (Cañadillas et al., 2020; Platis et al., 2021). The presented transect analysis also differs strongly from such studies in that it considers time-averages of different wind speeds and covers a very large extent with the stability and wind direction criterion only evaluated at the center of the transects. Added to that, modifications of dynamic stability by wind farms, which have previously been modelled with LES (Porté-Agel et al., 2014; Lu and Porté-Agel, 2015), could be strongly enhanced by the large, non-existent wind farms used in this study.*

**Comment 3**: Abstract, L11-13: "However, the long-term impact of wake losses in and between wind farms is mitigated by adopting next-generation,15 MW wind turbines instead of 5 MW turbines, as the layout-integrated, annual energy production (AEP) in the simulation increases by over 27% at the same capacity density." I am unsure what you mean by this sentence; I think the readers would, too. Would you reformulate?.

**Reply**: This was indeed a convoluted way to make clear that choosing 15 MW turbines over 5 MW turbines increased the layout efficiency. The use of the verb "mitigate" was therefore omitted and a more simplistic formulation was chosen. The next sentence has now been moved in front of this sentence to make the structure more logical. The adapted sentence is indicated in bold below

*The wind farm simulations indicate that for a typical capacity density and for SW-winds, inter-farm wakes can reduce the capacity factor at the inflow edge of wind farms from 59% to between 54% and 30% depending on the proximity, size and number of the upwind farms. The efficiency losses due to intra- and inter-farm wakes become larger with increasing capacity density as the layout-integrated, annual capacity factor varies between 51.8% and 38.2% over the considered range of 3.5 to 10 MW $km^{-2}$.* ***Also, the simulated efficiency of the wind farm layout is greatly impacted by switching from 5 MW turbines to next-generation, 15 MW turbines, as the annual energy production increases by over 27% at the same capacity density.***

**Comment 4**: Equation 2. In Latex, $PSS$ results in separated characters that look weird. Please use to fix this. The same applies to CF and other variables with more than one character throughout the manuscript.

**Reply**: We agree that this way of writing the variables did not fit the text well. Therefore, all mentions of variables were adapted accordingly in the manuscript and the equations. This was done throughout, to ensure consistency. This concerns equations (1)-(6) and the associated variable mentions throughout the text.

**Comment 5**: In figure 8 caption, indicate it is mean wind speed "deficit".

**Reply**: We have currently phrased the caption as "Relative deficit of the along-transect mean wind speed". Also in the label for the y-axis of the figure we have repeated this. We have tried to rephrase the caption slightly, but have eventually decided to keep the original formulation as it seemed to us the most readable.

---

## Author Response (AR3)

**Final author response for WES-2023-33**

February 7, 2024

**general reply**

We would like to thank again the editor and all of the reviewers for investing time and effort in reviewing our manuscript. This has had a substantial positive impact on the work and we are pleased to have the manuscript accepted!